# Antigen-Specific Antibody Design and Optimization with Diffusion-Based Generative Models for Protein Structures

**Shitong Luo**[1*], **Yufeng Su**[2*], **Xingang Peng**[3], **Sheng Wang**[4], **Jian Peng**[1,2], **Jianzhu Ma**[1,5]

[1] Helixon Research
[2] University of Illinois Urbana-Champaign
[3] School of Intelligence Science and Technology, Peking University
[4] Paul G. Allen School of Computer Science, University of Washington
[5] Institute for AI Industry Research, Tsinghua University

`luost@helixon.com,luost26@gmail.com`
`swang@cs.washington.edu,jianpeng@illinois.edu,majianzhu@tsinghua.edu.cn`

## Abstract

Antibodies are immune system proteins that protect the host by binding to specific antigens such as viruses and bacteria. The binding between antibodies and antigens is mainly determined by the complementarity-determining regions (CDR) of the antibodies. In this work, we develop a deep generative model that jointly models sequences and structures of CDRs based on diffusion probabilistic models and equivariant neural networks. Our method is the first deep learning-based method that generates antibodies explicitly targeting specific antigen structures and is one of the earliest diffusion probabilistic models for protein structures. The model is a "Swiss Army Knife" capable of sequence-structure co-design, sequence design for given backbone structures, and antibody optimization. We conduct extensive experiments to evaluate the quality of both sequences and structures of designed antibodies. We find that our model could yield competitive results in binding affinity measured by biophysical energy functions and other protein design metrics.

## 1 Introduction

Antibodies are important immune proteins generated during an immune response to recognize and neutralize the pathogen [Janeway et al., 2001]. As illustrated in Figure 1a, an antibody contains two heavy chains and two light chains, and their overall structure is similar. Six variable regions determine the specificity of an antibody to the antigens. They are called the Complementarity Determining Regions (CDRs), denoted as H1, H2, H3, L1, L2, and L3. Therefore, the most important step for developing effective therapeutic antibodies is to design CDRs that bind to the specific antigen [Presta, 1992, Akbar et al., 2022a].

Similar to other protein design tasks, the search space of CDRs is vast. A CDR sequence with $L$ amino acids has up to $20^L$ possible protein sequences. It is not feasible to test all the possible sequences using experimental approaches, so computational methods are needed. Traditional computational approaches rely on sampling protein sequences and structures from complex biophysical energy functions [Pantazes and Maranas, 2010, Lapidoth et al., 2015, Adolf-Bryfogle et al., 2018, Warszawski et al., 2019]. They are generally time-consuming and are prone to get trapped in local optima. Recently, various deep generative models have been developed to design antibodies [Saka et al., 2021, Akbar et al., 2022b, Jin et al., 2022]. Compared to conventional algorithms, deep generative models

---

[*]Equal contribution.

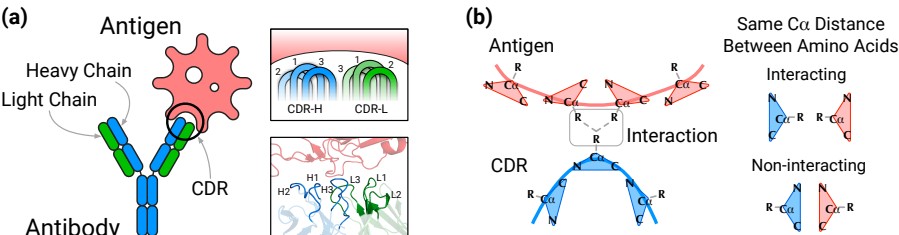

Figure 1: **(a)** Antibody-antigen complex structure and CDR structure. **(b)** The orientations of amino acids (represented by triangles) determine their side-chain orientations, which are key to inter-amino-acid interactions.

could directly capture higher-order interactions among amino acids on antibodies and antigens and generate antibodies more efficiently [Akbar et al., 2022a]. Recently, Jin et al. proposed a generative model for antibody structure-sequence co-design. Their model addresses two important computational challenges: First is how to model the intrinsic relation between CDR sequences and 3D structures, and second is how to model the distribution of CDRs conditional on the rest of the antibody sequence. However, there is still a large gap to fill before generative models become practical for antibody design.

Here, we identify another three challenges for antibody sequence-structure co-design. *First*, the model should be *explicitly conditional on the 3D structures of the antigen* and generate CDRs that fit the antigen structure in the 3D space. This is indispensable for the model to generalize to new antigens. *Second*, the interactions between amino acids are mainly determined by side-chains which are groups of atoms stretching out from the protein backbone (Figure 1b) [Liljas et al., 2016]. Therefore, the model should be able to consider both the *position* and *orientation* of amino acids. *Third*, in drug discovery, pharmacologists collect multiple initial antibodies either from humanized mice or patients [Presta, 1992, Barlow et al., 2018, Warszawski et al., 2019]. Therefore, instead of *de novo* design, the model should be applicable to another realistic scenario: optimizing a particular antibody to increase the binding affinity to the antigen. To the best of our knowledge, no previous machine learning model satisfies all of the above design principles.

To address these challenges, we propose a diffusion-based generative model [Sohl-Dickstein et al., 2015, Song and Ermon, 2019, Ho et al., 2020, Yang et al., 2022] capable of jointly sampling antibody CDR sequences and structures. More importantly, the joint distribution of a CDR sequence and its structure is *directly conditional on antigen structures*. Given a protein complex consisting of an *antigen* and an *antibody framework* as input[2] (as illustrated in Figure 2), we first initialize the CDR with an arbitrary sequence, positions, and orientations. The diffusion model first aggregates information from the antigen and the antibody framework. Then, it iteratively updates the amino

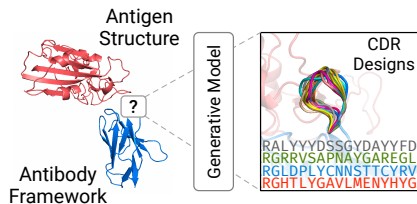

Figure 2: The task in this work is to design CDRs for a given antigen structure and an antibody framework.

acid type, position, and orientation of each amino acid on CDRs. In the last step, we reconstruct the CDR structure at the atom level using side-chain packing algorithms based on the predicted orientations [Alford et al., 2017]. From the perspective of model capability, one of the most important reasons for us to choose the diffusion-based model over other generative models such as generative adversarial networks [Goodfellow et al., 2014] and variational auto-encoders [Kingma and Welling, 2013] is that it generates CDR candidates iteratively in the sequence-structure space so that we can interfere and impose constraints on the sampling process to support a broader range of design tasks.

We summarize our contributions as follows:

- We propose the first deep learning models to perform antibody sequence-structure design by considering the 3D structures of the antigen.
- In our model, we not only design protein sequences and coordinates but also side-chain orientations (represented as $SO(3)$ element) of each amino acid. It is the first deep learning

---

[2]The structure of the antigen-antibody framework can be obtained either from existing antigen-antibody structure or by docking an initial antibody to the target antigen.

model that could achieve atomic-resolution antibody design and is equivariant to rotation and translation.

- We show that our model can be applied to a wide range of antibody design tasks, including sequence-structure co-design, fix-backbone CDR design, and antibody optimization.

## 2 Related Work

**Computational Antibody Design** Conventional computational approaches are mainly based on sampling algorithms over hand-crafted and statistical energy functions and iteratively modify protein sequences and structures [Adolf-Bryfogle et al., 2018, Lapidoth et al., 2015, Warszawski et al., 2019, Pantazes and Maranas, 2010, Ruffolo et al., 2021]. These methods are inefficient and prone to getting stuck at local optima due to the rough energy landscape. In recent years, deep learning methods have shown potential in antibody design by using language models to generate protein sequences [Alley et al., 2019, Shin et al., 2021, Saka et al., 2021, Akbar et al., 2022b]. Although much more efficient, the sequence-based methods can only generate new antibodies based on previously observed antibodies but can hardly generate antibodies for specific antigen structures.

Jin et al. proposed the first CDR sequence-structure co-design deep generative model which focuses on designing antibodies to neutralize SARS-CoV-2. *It relies on an additional antigen-specific predictor to predict the neutralization of the designed antibodies, which is not generalizable to arbitrary antigens.* In comparison to their model, we explicitly model the 3D structure of an antigen, opening the door to generalizing the prediction to unseen antigens with solved 3D structures. Another advantage of our model is that we consider not only backbone atom coordinates but also the orientation of amino acids. The orientation is critical to protein-protein interactions as most of the atoms interacting between antibodies and antigens are in the side-chain [Liljas et al., 2016] (as illustrated in Figure 1b). Lastly, the model proposed by Jin et al. is not equivariant by construction, which is fundamental in molecular modeling.

**Protein Structure Prediction** Protein structure prediction algorithms take protein sequences and Multiple Sequence Alignments (MSAs) as input and translate them to 3D structures [Jumper et al., 2021, Baek et al., 2021, Yang et al., 2020]. Accurate protein structure prediction models predict not only the position of amino acids but also their orientation [Jumper et al., 2021, Yang et al., 2020]. The orientation of amino acids determines the direction in which its side chain stretches, so it is indispensable for reconstructing full-atom structures. AlphaFold2 [Jumper et al., 2021] predicts per-amino-acid orientations in an iterative fashion, similar to our proposed model. However, it is not generative, unable to efficiently sample diverse structures for protein design. Recently, based on prior protein structure prediction algorithms, methods for predicting antibody CDR structures have emerged [Ruffolo et al., 2022b,a], but they are not able to design CDR sequences.

**Diffusion-Based Generative Models** Diffusion probabilistic models learn to generate data via denoising samples from a prior distribution [Sohl-Dickstein et al., 2015, Song and Ermon, 2019, Ho et al., 2020]. Recently, progress has been made in developing equivariant diffusion models for molecular 3D structures [Shi et al., 2021, Hoogeboom et al., 2022, Jing et al., 2022, Xu et al., 2022]. Atoms in a molecule do not have natural orientations, so the generation process differs from generating protein structures. Diffusion models have also been extended to non-Euclidean data, such as data in the Riemannian manifolds [Leach et al., 2022, De Bortoli et al., 2022]. These models are relevant to modeling orientations which are represented by elements in SO(3). In addition, diffusion models can also be used to generate discrete categorical data [Hoogeboom et al., 2021, Austin et al., 2021]. Concurrently with this work, various diffusion probabilistic models have been developed for proteins [Anand and Achim, 2022, Trippe et al., 2022, Wu et al., 2022].

## 3 Methods

This section is organized as follows: Section 3.1 introduces notations used throughout the paper and formally states the problem. Section 3.2 formulates the diffusion process for modeling antibodies. Section 3.3 introduces details about the neural network parameterization for the diffusion processes. Section 3.4 presents sampling algorithms for various antibody design tasks.

## 3.1 Definitions and Notations

An amino acid in a protein complex can be represented by its type, $C_\alpha$ atom coordinate, and the orientation, denoted as $s_i \in \{\texttt{ACDEFGHIKLMNPQRSTVWY}\}, \boldsymbol{x}_i \in \mathbb{R}^3, \boldsymbol{O}_i \in \mathrm{SO}(3)$, respectively. Here $i = 1 \ldots N$, and $N$ is the number of amino acids in the protein complex[3].

In this work, we assume the antigen structure and the antibody framework is given (Figure 2), and we focus on designing CDRs on the antibody framework. Assume the CDR to be generated has $m$ amino acids with index from $l+1$ to $l+m$. They are denoted as $\mathcal{R} = \{(s_j, \mathbf{x}_j, \boldsymbol{O}_j) \mid j = l+1, \ldots, l+m\}$. Formally, our goal is to jointly model the distribution of $\mathcal{R}$ given the structure of the antibody-antigen complex $\mathcal{C} = \{(s_i, \boldsymbol{x}_i, \boldsymbol{O}_i) \mid i \in \{1 \ldots N\} \backslash \{l+1, \ldots, l+m\}\}$.

## 3.2 Diffusion Processes

A diffusion probabilistic model defines two Markov chains of diffusion processes. The forward diffusion process gradually adds noise to the data until the data distribution approximately reaches the prior distribution. The generative diffusion process starts from the prior distribution and iteratively transforms it to the desired distribution. Training the model relies on the forward diffusion process to simulate the noisy data. Let $(s_j^t, \mathbf{x}_j^t, \mathbf{O}_j^t)$ denote the intermediate state of amino acid $j$ at time step $t$. $\mathcal{R}^t = \{s_j^t, \mathbf{x}_j^t, \mathbf{O}_j^t\}_{j=l+1}^{l+m}$ represents the sequence and structure sampled at step $t$. $t = 0$ represents the state of real data (observed sequences and structures of CDRs) and $t = T$ represents samples from the prior distribution. Forward diffusion goes from $t = 0$ to $T$, and generative diffusion proceeds in the opposite way. The diffusion processes for amino acid types $s_j^t$, coordinates $\mathbf{x}_j^t$, and orientations $\mathbf{O}_j^t$ are defined as follows:

**Multinomial Diffusion for Amino Acid Types** The forward diffusion process for amino acid types is based on the multinomial distribution defined as follows [Hoogeboom et al., 2021]:

$$q(s_j^t | s_j^{t-1}) = \mathrm{Multinomial}\left((1 - \beta_{\text{type}}^t) \cdot \texttt{onehot}(s_j^{t-1}) + \beta_{\text{type}}^t \cdot \frac{1}{20} \cdot \mathbf{1}\right), \tag{1}$$

where $\texttt{onehot}$ represents a function that converts amino acid type to a 20-dimensional one-hot vector and $\mathbf{1}$ is an all-one vector. $\beta_{\text{type}}^t$ is the probability of resampling another amino acid over 20 types uniformly. When $t \to T$, $\beta_{\text{type}}^t$ is set close to 1 and the distribution is closer to the uniform distribution. The following probability density provides an efficient way to perturb $s_j^0$ for timestep $t$ during training [Hoogeboom et al., 2021]:

$$q(s_j^t | s_j^0) = \mathrm{Multinomial}\left(\bar{\alpha}_{\text{type}}^t \cdot \texttt{onehot}(s_j^0) + (1 - \bar{\alpha}_{\text{type}}^t) \cdot \frac{1}{20} \cdot \mathbf{1}\right), \tag{2}$$

where $\bar{\alpha}_{\text{type}}^t = \prod_{\tau=1}^t (1 - \beta_{\text{type}}^\tau)$.

The generative diffusion process is defined as:

$$p(s_j^{t-1} | \mathcal{R}^t, \mathcal{C}) = \mathrm{Multinomial}\left(F(\mathcal{R}^t, \mathcal{C})[j]\right), \tag{3}$$

where $F(\cdot)[j]$ is a neural network model taking the structure context (antigen and antibody framework) and the CDR state from the previous step as input and predicts the probability of the amino acid type for the $j$-th amino acid on the CDR. Note that, different from the forward diffusion process, the generative diffusion process must rely on the structure context $\mathcal{C}$ and the CDR state of the previous step including positions and orientations. The main difference between these two processes is that the forward diffusion process adds noise to data so it is irrelevant to data or contexts but the generative diffusion process depends on the given condition and full observation of the previous step. The generative diffusion process needs to approximate the posterior $q(s_j^{t-1} | s_j^t, s_j^0)$ derived from Eq.1 and Eq.2 to denoise. Therefore, the objective of training the generative diffusion process for amino acid types is to minimize the expected KL divergence between Eq.3 and the posterior distribution:

$$L_{\text{type}}^t = \mathbb{E}_{\mathcal{R}^t \sim p}\left[\frac{1}{m} \sum_j D_{\mathrm{KL}}\left(q(s_j^{t-1} | s_j^t, s_j^0) \middle\| p(s_j^{t-1} | \mathcal{R}^t, \mathcal{C})\right)\right]. \tag{4}$$

---

[3]Note that a protein complex contains more than one chain, so $N$ is not the length of one protein but is the sum of the lengths of all chains in the complex.

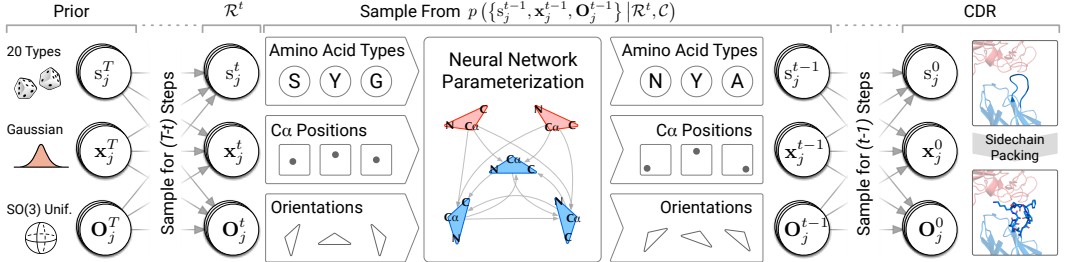

Figure 3: Illustration of the generative diffusion process. At each step, the network takes the current CDR state as input and parameterizes the distribution of the CDR's sequences, positions, and orientations for the next step. In the end, full-atom structures are constructed by the side-chain packing algorithm.

**Diffusion for $C_\alpha$ Coordinates** As the coordinate of an atom could be an arbitrary value, we scale and shift the coordinates of the whole structure such that the distribution of atom coordinates roughly match the standard normal distribution. We define the forward diffusion for the normalized $C_\alpha$ coordinate $\mathbf{x}_j$ as follows:

$$q\left(\mathbf{x}_j^t \mid \mathbf{x}_j^{t-1}\right) = \mathcal{N}\left(\mathbf{x}_j^t \middle| \sqrt{1 - \beta_{\text{pos}}^t} \cdot \mathbf{x}_j^{t-1}, \beta_{\text{pos}}^t \boldsymbol{I}\right), \tag{5}$$

$$q\left(\mathbf{x}_j^t \mid \mathbf{x}_j^0\right) = \mathcal{N}\left(\mathbf{x}_j^t \middle| \sqrt{\bar{\alpha}_{\text{pos}}^0} \cdot \mathbf{x}_j^0, (1 - \bar{\alpha}_{\text{pos}}^0)\boldsymbol{I}\right), \tag{6}$$

where $\beta_{\text{pos}}^t$ controls the rate of diffusion and its value increases from 0 to 1 as time step goes from 0 to $t$, and $\bar{\alpha}_{\text{pos}}^t = \prod_{\tau=1}^t (1 - \beta_{\text{pos}}^\tau)$. Using the reparameterization trick proposed by Ho et al., the generative diffusion process is defined as:

$$p\left(\mathbf{x}_j^{t-1} \middle| \mathcal{R}^t, \mathcal{C}\right) = \mathcal{N}\left(\mathbf{x}_j^{t-1} \middle| \boldsymbol{\mu}_p\left(\mathcal{R}^t, \mathcal{C}\right), \beta_{\text{pos}}^t \boldsymbol{I}\right), \tag{7}$$

$$\boldsymbol{\mu}_p\left(\mathcal{R}^t, \mathcal{C}\right) = \frac{1}{\sqrt{\alpha_{\text{pos}}^t}}\left(\mathbf{x}_j^t - \frac{\beta_{\text{pos}}^t}{\sqrt{1 - \bar{\alpha}_{\text{pos}}^t}} G(\mathcal{R}^t, \mathcal{C})[j]\right). \tag{8}$$

Here, $G(\cdot)[j]$ is a neural network that predicts the standard Gaussian noise $\epsilon_j \sim \mathcal{N}(\mathbf{0}, \boldsymbol{I})$ added to $\sqrt{\bar{\alpha}_{\text{pos}}^0}\mathbf{x}_j^0$ (scaled coordinate of amino acid $j$) based on the reparameterization of Eq.6: $\mathbf{x}_j^t = \sqrt{\bar{\alpha}_{\text{pos}}^0}\mathbf{x}_j^0 + \sqrt{1 - \bar{\alpha}_{\text{pos}}^0}\epsilon_j$. The objective function of training the generative process is the expected MSE between $G$ and $\epsilon_j$, which is simplified from aligning distribution $p$ to the posterior $q(\mathbf{x}_j^{t-1} \mid \mathbf{x}_j^t, \mathbf{x}_j^0)$ [Ho et al., 2020]:

$$L_{\text{pos}}^t = \mathbb{E}\left[\frac{1}{m} \sum_j \left\| \epsilon_j - G(\mathcal{R}^t, \mathcal{C}) \right\|^2\right]. \tag{9}$$

**SO(3) Denoising for Amino Acid Orientations** We empirically formulate an iterative perturb-denoise scheme for learning and generating amino acid orientations represented by SO(3) elements [Leach et al., 2022]. Note that we do not use the term *diffusion* because the formulation does not strictly follow the framework of diffusion probabilistic models though the overall principle is the same. Similar to the typical diffusion process, the distribution of orientations perturbed for $t$ steps is defined as, according to Leach et al. [2022]:

$$q\left(\mathbf{O}_j^t \mid \mathbf{O}_j^0\right) = \mathcal{IG}_{\text{SO(3)}}\left(\mathbf{O}_j^t \middle| \text{ScaleRot}\left(\sqrt{\bar{\alpha}_{\text{ori}}^t}, \mathbf{O}_j^0\right), 1 - \bar{\alpha}_{\text{ori}}^t\right). \tag{10}$$

$\mathcal{IG}_{\text{SO(3)}}$ denotes the isotropic Gaussian distribution on SO(3) parameterized by a mean rotation and a scalar variance [Leach et al., 2022, Matthies et al., 1970, Nikolayev and Savyolov, 1970]. ScaleRot modifies the rotation matrix by scaling its rotation angle with the rotation axis fixed [Gallier and Xu,

2003]. $\bar{\alpha}_{\text{ori}}^t = \prod_{\tau=1}^t (1 - \beta_{\text{ori}}^\tau)$, where $\beta_{\text{ori}}^t$ is the variance increases with the step $t$. The conditional distribution used for the generation process of orientations is defined as:

$$p\left(\mathbf{O}_j^{t-1}\middle|\mathcal{R}^t,\mathcal{C}\right) = \mathcal{IG}_{\text{SO(3)}}\left(\mathbf{O}_j^{t-1}\middle|H(\mathcal{R}^t,\mathcal{C})[j], \beta_{\text{ori}}^t\right), \tag{11}$$

where $H(\cdot)[j]$ is a neural network that denoises the orientation and outputs the denoised orientation matrix of amino acid $j$. Training the conditional distribution requires aligning the predicted orientation from $H(\cdot)$ to the real orientation. Hence, we formulate the training object that minimizes the expected discrepancy measured by the inner product between the real and the predicted orientation matrices:

$$L_{\text{ori}}^t = \mathbb{E}\left[\frac{1}{m}\sum_j \left\|(\mathbf{O}_j^0)^\intercal \widehat{\mathbf{O}}_j^{t-1} - \boldsymbol{I}\right\|_F^2\right], \tag{12}$$

where $\widehat{\mathbf{O}}_j^{t-1} = H(\cdot)[j]$ is the predicted orientation for amino acid $j$.

**The Overall Training Objective**   By summing Eq.4, 9, and 12 and taking the expectation w.r.t. $t$, we obtain the final training objective function:

$$L = \mathbb{E}_{t\sim\text{Uniform}(1...T)}\left[L_{\text{type}}^t + L_{\text{pos}}^t + L_{\text{ori}}^t\right]. \tag{13}$$

To train the model, we first sample a time step $t$ and then sample noisy states $\{s_j^t, \mathbf{x}_j^t, \mathbf{O}_j^t\}_{j=l+1}^{l+m} \sim p$ by adding noise to the training sample using the diffusion process defined by Eq.2, 6, and 10. We compute the loss using the noisy data and backpropagate the loss to update model parameters.

## 3.3   Parameterization with Neural Networks

In this section, we briefly introduce the neural network architectures used in different components of the diffusion process. The purpose of the networks is to encode the CDR state at a time step $t$ along with the context structure: $\{s_j^t, \mathbf{x}_j^t, \mathbf{O}_j^t\}_{j=l+1}^{l+m} \cup \{s_i^t, \boldsymbol{x}_i^t, \boldsymbol{O}_i^t\}_{i=\{1...N\}\setminus\{l+1...l+m\}}$, and then denoises the CDR amino acid types ($F$), positions ($G$), and orientations ($H$).

First, we adopt Multiple Layer Perceptrons (MLPs) to generate embeddings for single and pairs of amino acids. The single amino-acid embedding MLP creates vector $\boldsymbol{e}_i$ for amino acid $i$, which encodes the information of amino acid types, torsional angles, and 3D coordinates of all the heavy atoms. The pairwise embedding MLP encodes the Euclidean distances and dihedral angles between amino acid $i$ and $j$ to feature vectors $\boldsymbol{z}_{ij}$. We adopt IPA [Jumper et al., 2021], an orientation-aware roto-translation invariant network to transform $\boldsymbol{e}_i$ and $\boldsymbol{z}_{ij}$ into hidden representations $\boldsymbol{h}_i$, which aims to represent the amino acid itself and its environment. Next, the representations are fed to three different MLPs to denoise the amino acid types, 3D positions, and orientations of the CDR, respectively.

In particular, the MLP for denoising amino acid types outputs a 20-dimensional vector representing the posterior probabilities. The MLP for denoising $\text{C}_\alpha$ coordinates predicts the scaled change of the coordinate in terms of the current orientation of the amino acid. As the coordinate deviation is calculated in the local frame, we left-multiply it by the orientation matrix and transform it back to the global frame. Formally, this can be expressed as $\hat{\epsilon}_j = \mathbf{O}_j^t \text{MLP}_G(\boldsymbol{h}_j)$. Predicting coordinate deviations in the local frame and projecting it to the global frame ensures the equivariance of the prediction, as when the entire 3D structure rotates by a particular angle, the coordinate deviations also rotate by the same angle. The MLP for denoising orientations first predicts a $so(3)$ vector [Gallier and Xu, 2003]. The vector is converted to a rotation matrix $M_j \in \text{SO}(3)$ right-multiplied to the orientation to produce a new mean orientation for the next generative step: $\widehat{\mathbf{O}}_j^{t-1} \leftarrow \mathbf{O}_j^t M_j$. The proposed networks are equivariant to the rotation and translation of the overall structure:

**Proposition 1.** *For any proper rotation matrix $\boldsymbol{R} \in \text{SO}(3)$ and any 3D vector $\boldsymbol{r} \in \mathbb{R}^3$ (rigid transformation $(\boldsymbol{R}, \boldsymbol{r}) \in \text{SE}(3)$), $F$, $G$ and $H$ satisfy the following equivariance properties:*

$$F(\boldsymbol{R}\mathcal{R}^t + \boldsymbol{r}, \boldsymbol{R}\mathcal{C} + \boldsymbol{r}) = F(\mathcal{R}^t, \mathcal{C}), \tag{14}$$

$$G(\boldsymbol{R}\mathcal{R}^t + \boldsymbol{r}, \boldsymbol{R}\mathcal{C} + \boldsymbol{r}) = \boldsymbol{R}G(\mathcal{R}^t, \mathcal{C}), \tag{15}$$

$$H(\boldsymbol{R}\mathcal{R}^t + \boldsymbol{r}, \boldsymbol{R}\mathcal{C} + \boldsymbol{r}) = \boldsymbol{R}H(\mathcal{R}^t, \mathcal{C}), \tag{16}$$

*where $\boldsymbol{R}\mathcal{R}^t + \boldsymbol{r} := \{s_j^t, \mathbf{x}_j^t + \boldsymbol{r}, \boldsymbol{R}\mathbf{O}_j^t\}_{j=l+1}^{l+m}$ and $\boldsymbol{R}\mathcal{C} + \boldsymbol{r} := \{s_i, \boldsymbol{x}_i + \boldsymbol{r}, \boldsymbol{R}\boldsymbol{O}_i\}_{i\in\{1...N\}\setminus\{l+1,...,l+m\}}$ denote the rotated and translated structure. Note that $F$, $G$, and $H$ are not single MLPs. Each of them includes the shared encoder and a specific MLP.*

## 3.4 Sampling Algorithms

The sampling algorithm first samples amino acid types from the uniform distribution over 20 classes: $s_j^T \sim \text{Uniform}(20)$, $C_\alpha$ positions from the standard normal distribution: $\mathbf{x}_j^T \sim \mathcal{N}(\mathbf{0}, \mathbf{I}_3)$, and orientations from the uniform distribution over SO(3): $\mathbf{O}_j^T \sim \text{Uniform}(\text{SO}(3))$. Note that we normalize the coordinates of the structure in the same way as training such that $C_\alpha$ positions in the CDR roughly follow the standard normal distribution. Next, we iteratively sample sequences and structures from the generative diffusion kernel by denoising amino acid types, $C_\alpha$ coordinates, and orientations until $t = 0$. To build a full atom 3D structure, we construct the coordinates of N, $C_\alpha$, C, O, and side-chain $C_\beta$ (except glycine that does not have $C_\beta$) according to their ideal local coordinates relative to the $C_\alpha$ position and orientation of each amino acid [Engh and Huber, 2012]. Based on the five reconstructed atoms, the rest of the side-chain atoms are constructed using the side-chain packing function implemented in Rosetta [Alford et al., 2017]. In the end, we adopt the AMBER99 force field [Lindorff-Larsen et al., 2010] in OpenMM [Eastman et al., 2017] to refine the full atom structure.

In addition to the joint design of sequences and structures, we can constrain partial states for other design tasks. For example, by fixing the backbone structure (positions and orientations) and sampling only sequences, we can do **fix-backbone sequence design**. Another usage is to **optimize an existing antibody**. Specifically, we first add noise to the existing antibody for $t$ steps and denoise the perturbed antibody sequence starting from the $t$-th step of the generative diffusion process.

# 4 Experiments

We present the application of our model, named **DiffAb**[4], in three antibody design tasks: sequence-structure co-design (Section 4.1), antibody sequence design based on antibody backbones (Section 4.2), and antibody optimization (Section 4.3). In Section 4.4, we show how to use our model without known antibody frameworks bound to the antigen.

## 4.1 Sequence-Structure Co-design

The dataset for training the model is derived from the SAbDab database[Dunbar et al., 2014]. We first remove structures whose resolution is worse than 4Å and discard antibodies targeting non-protein antigens. We cluster antibodies in the database according to CDR-H3 sequences at 50% sequence identity. We manually select five clusters as the test set, containing 19 antibody-antigen complexes in total. The test set includes antigens from several well-known pathogens including SARS-CoV-2, MERS, influenza, and so on. Structures in the remaining clusters are used for training.

To evaluate the performance, we remove the original CDR from the antibody-antigen complex in the test set and sample both the sequence and structure of the removed region. We set the length of the CDR to be identical to the length of the original CDR for simplicity. In practice, one can enumerate different lengths of CDRs. We compare our model to RosettaAntibodyDesign (RAbD) [Adolf-Bryfogle et al., 2018], an antibody design software based on Rosetta energy functions. For each model, we draw 100 samples for each CDR. Both the original structures and designed structures from different methods are refined by OpenMM and Rosetta.

We use the following metrics to evaluate designed antibodies: (1) IMP: is the percentage of designed CDRs with lower (better) binding energy ($\Delta G$) than the original CDR. The binding energy is calculated by `InterfaceAnalyzer` in the Rosetta software package [Alford et al., 2017]. (2) RMSD: is the $C_\alpha$ root-mean-square deviation (RMSD) between the generated structure and the original structure with *only antibody frameworks aligned*. (3) AAR: is the amino acid recovery rate measured by the sequence identity between the reference CDR sequences and the generated sequences [Adolf-Bryfogle et al., 2018]. Note that different from Jin et al. [2022], we do not use neutralization prediction models because they are sequence-based and are specified to a limited class of antigens, which deviates from our goal of developing a general antibody design model.

Table 1 shows that our model (DiffAb) recovers CDR sequences more accurately than RAbD (higher AAR). The RMSDs of CDRs generated by DiffAb are higher in CDR-H3, which indicates that our generated samples are more diverse structurally. The IMP score of DiffAb is on par with RAbD in

---

[4]Code and data are available at `https://github.com/luost26/diffab`.

Table 1: Evaluation of the generated antibody CDRs (sequence-structure co-design) by RAbD and our DiffAb model.

| CDR | Method | AAR | RMSD | IMP | CDR | Method | AAR | RMSD | IMP |
|-----|--------|-----|------|-----|-----|--------|-----|------|-----|
| H1 | RAbD | 22.85% | 2.261Å | 43.88% | L1 | RAbD | 34.27% | 1.204Å | 46.81% |
| | DiffAb | 65.75% | 1.188Å | 53.63% | | DiffAb | 56.67% | 1.388Å | 45.58% |
| H2 | RAbD | 25.50% | 1.641Å | 53.50% | L2 | RAbD | 26.30% | 1.767Å | 56.94% |
| | DiffAb | 49.31% | 1.076Å | 29.84% | | DiffAb | 59.32% | 1.373Å | 49.95% |
| H3 | RAbD | 22.14% | 2.900Å | 23.25% | L3 | RAbD | 20.73% | 1.624Å | 55.63% |
| | DiffAb | 26.78% | 3.597Å | 23.63% | | DiffAb | 46.47% | 1.627Å | 47.32% |

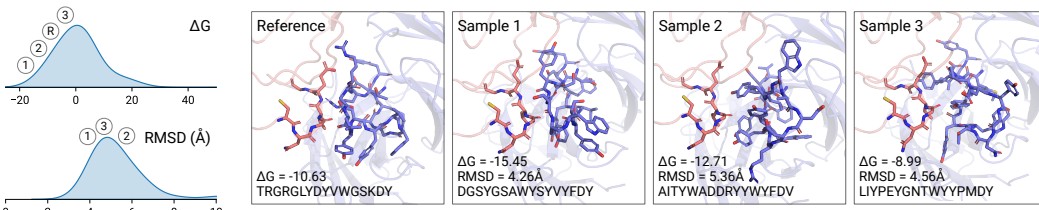

Figure 4: Examples of CDR-H3 designed by the sequence-structure co-design method and the distribution of their interaction energy and RMSD. The antigen-antibody template is derived from PDB:7chf, where the antigen is SARS-CoV-2 RBD. Sample 1 has better complementarity to the antigen while Sample 3 fits the antigen worse. This could explain their difference in the binding energy ($\Delta G$).

CDR-H3, and lower in other CDRs. However, it should be noted that RAbD optimizes the Rosetta energy function, which is also used for evaluation. Our model achieves reasonably good binding energy without explicit supervision signal from Rosetta energy functions. Figure 4 presents three generated examples of CDR-H3 targeting SARS-CoV-2 RBD. Sample 1 has the lowest binding energy and it can be observed that it has better complementarity to the antigen. The binding energy of Sample 3 is higher than the original one and visually, the shape of the CDR does not fit the antigen well.

### 4.2 Fix-Backbone Sequence Design and Structure Prediction

In this setting, the backbone structure of CDRs is given and we only need to design the CDR sequence, which transforms the task into a constrained sampling problem. Fix-backbone design is a common setting in the area of protein design [Ingraham et al., 2019, Hsu et al., 2022, Anishchenko et al., 2021, Strokach et al., 2020, Tischer et al., 2020]. For this task, we consider **FixBB**, a Rosetta-based sequence design software given CDR backbone structure, as the baseline. We use the AAR metric introduced in Section 4.1 to evaluate the designed CDRs.

As shown in Table 2, our model achieves better AAR in all the CDRs. This shows that our model is also powerful in modeling the conditional probability of sequences given backbone structures. Admittedly, the training data is clustered only by CDR-H3 sequences, so the model might have seen other CDRs in the test set during training, leading to even higher AAR. However, we believe this is not an issue as CDRs other than H3 are generally conserved and contribute less to the specificity Xu and Davis [2000].

Our model can predict CDR structures by fixing the sequence. Table 3 shows that it accurately predicts the structure of CDR H1, H2, L1, L2, and L3 (RMSD ≤ 1.5Å). The accuracy of CDR-H3 prediction is lower due to the high variability. Figure 5a separately shows the accuracy of different CDR-H3 lengths. The prediction is generally more accurate for shorter ones. When the CDR-H3 contains more than 10 amino acids, the prediction accuracy drops.

Table 2: Comparison of FixBB and DiffAb in terms of amino acid recovery (AAR) in the fix-backbone CDR design task. DiffAb achieves higher AAR. The AAR of DiffAb on CDR-H3 is lower than other CDRs since H3 is much more versatile.

| CDR | Method | AAR | CDR | Method | AAR |
|---|---|---|---|---|---|
| H1 | FixBB | 37.14% | L1 | FixBB | 33.80% |
| | DiffAb | 87.83% | | DiffAb | 86.63% |
| H2 | FixBB | 43.08% | L2 | FixBB | 28.54% |
| | DiffAb | 79.70% | | DiffAb | 88.91% |
| H3 | FixBB | 30.74% | L3 | FixBB | 17.92% |
| | DiffAb | 59.48% | | DiffAb | 78.69% |

Table 3: The accuracy of CDR structures predicted by DiffAb in RMSD.

| CDR | RMSD |
|---|---|
| H1 | 0.901Å |
| H2 | 1.044Å |
| H3 | 3.246Å |
| L1 | 1.365Å |
| L2 | 1.321Å |
| L3 | 1.492Å |

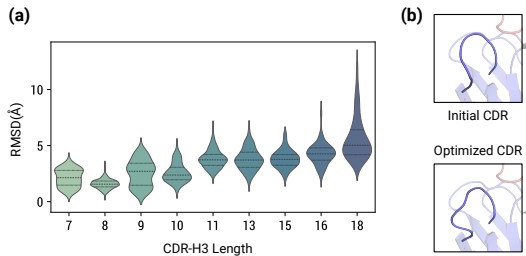 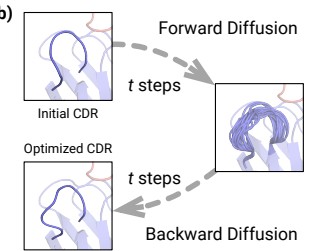 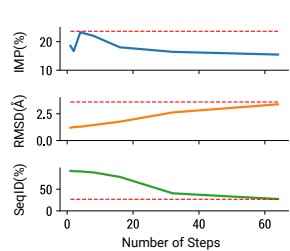

Figure 5: **(a)** RMSD of predicted CDR-H3 structures grouped by lengths. **(b)** The antibody optimization algorithm first perturbs the initial CDR for $t$ steps using the forward diffusion process and then denoises it by the backward diffusion process into the optimized CDR. **(c)** IMP, RMSD, and SeqID of the CDRs optimized with different numbers of steps. Dashed lines represent the results of *de novo* design. When $t = 4$, the optimized CDRs reach an IMP score close to *de novo* CDRs but remain structurally similar to the original one.

## 4.3 Antibody Optimization

We use our model to optimize existing antibodies which is another common pharmaceutical application. To optimize an antibody, we *first perturb the CDR sequence and structure* for $t$ steps using the forward diffusion process. Then, we denoise the sequences starting from the $(T - t)$-th step ($t$ steps remaining) of the generative diffusion process and obtain a set of optimized antibodies. This process is illustrated in Figure 5b. We optimize CDR-H3 of the antibodies in the test set with various $t$ values. For each antibody and $t$, we perturb the CDR independently 100 times and collect 100 optimized CDRs different from the original CDR. We report the percentage of optimized antibodies with improved binding energy (IMP), RMSD, and sequence identity (SeqID) of the optimized CDR in comparison to the original antibody. We also compare the optimized antibodies with the *de novo* ($t = T = 100$) designed antibodies introduced in Section 4.1. As shown in Table 4 and Figure 5c, the optimization method could

Table 4: Evaluation of optimized CDR-H3s with different numbers of optimization steps. In contrast to redesigning the CDR, the optimization method can improve binding energy while *keeping the optimized CDR similar to the original one*. Figure 5c shows the line plot of the results.

| $t$ | IMP | RMSD | SeqID |
|---|---|---|---|
| 1 | 18.52% | 1.194Å | 92.42% |
| 2 | 16.67% | 1.252Å | 91.61% |
| 4 | 23.29% | 1.290Å | 91.16% |
| 8 | 22.01% | 1.447Å | 88.78% |
| 16 | 18.02% | 1.759Å | 78.43% |
| 32 | 16.43% | 2.623Å | 40.58% |
| 64 | 15.47% | 3.380Å | 27.30% |
| $T$ | 23.63% | 3.597Å | 26.78% |

produce antibodies with improved binding energy measured by the Rosetta energy function. In contrast to redesigning CDRs, optimization improves binding energy while keeping the optimized CDR similar to the original one, which is desired in many practical applications.

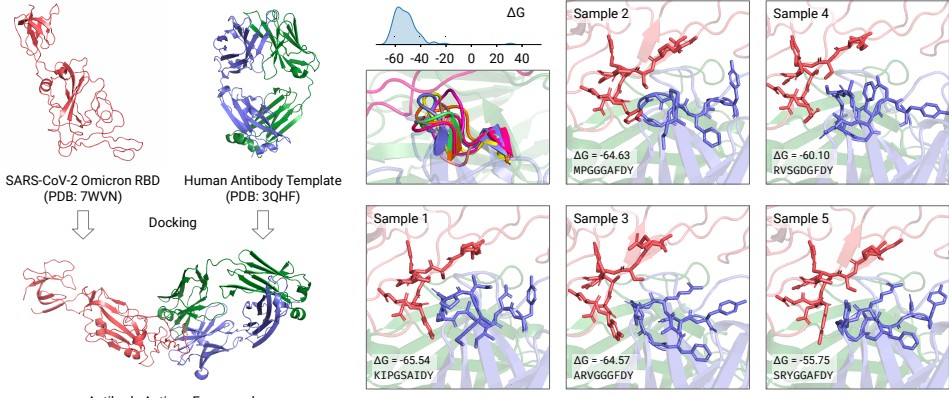

Figure 6: A human antibody framework docked to SARS-CoV-2 Omicron RBD using HDOCK. CDR-H3s are designed based on the docking structure.

## 4.4 Design Without Bound Antibody Frameworks

In the last experiment, we consider designing antibodies without a known binding pose against the antigen, a more general and challenging setting. We show that this challenging task could be achieved with docking software. Specifically, we create an *antibody template* from an existing antibody structure by removing its CDR-H3. This is because CDR-H3 is the most variable one and accounts for most of the specificity, while other CDRs are much more conserved [Xu and Davis, 2000]. Next, we use HDOCK [Yan et al., 2017] to dock the antibody template to the target antigen to produce the antibody-antigen complex. In this way, the problem reduces to the original problem so we can adapt our model to design the CDR-H3 sequence and structure and re-design other CDRs. We demonstrate using this method to design antibodies for the SARS-CoV-2 Omicron RDB structure (PDB: 7wvn, residue A322-A590, the structure is not bound to any antibodies). The antibody template is derived from a human antibody against influenza (PDB: 3qhf). Figure 6 shows the docking structure, five designed CDR-H3s, and the binding energy distribution. It is hard to confidently conclude that the generated antibodies are effective without a reference antibody. However, according to the binding energy distribution, we can still say the generated antibodies are at least reasonable.

## 5 Conclusions and Limitations

In this work, we propose a diffusion-based generative model for antibody design. Our model is capable of a wide range of antibody design tasks and can achieve competitive performance. One main limitation of this work is that it relies on an antibody framework bound to the target antigen. Therefore, we leave it for future work to design an effective model for generating antibodies without bound structures. Another limitation is that it remains unclear whether the generated antibodies can be produced in the wet lab and actually binds to the target. More efforts are needed to design a biologically effective antibody.

## Acknowledgments and Disclosure of Funding

Supported by National Key R&D Program of China No. 2021YFF1201600.

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
