# Appendix

## A  Diffusion Processes

### A.1  Posteriors

**Posterior of Amino Acid Types**   The generative diffusion kernel for amino acid types $p(s_j^{t-1}|\mathcal{R}^t, \mathcal{C})$ (Eq.3) should align to the posterior $q(s_j^{t-1}|s_j^t, s_j^0)$. It can be derived from Eq.1 and Eq.2 [Hoogeboom et al., 2021]:

$$q(s_j^{t-1}|s_j^t, s_j^0) = \text{Multinomial}\left(\left[\alpha_{\text{type}}^t \cdot \texttt{onehot}(s_j^t) + (1 - \alpha_{\text{type}}^t) \cdot \frac{1}{20} \cdot \mathbf{1}\right] \odot \right.$$
$$\left.\left[\bar{\alpha}_{\text{type}}^{t-1} \cdot \texttt{onehot}(s_j^0) + (1 - \bar{\alpha}_{\text{type}}^{t-1}) \cdot \frac{1}{20} \cdot \mathbf{1}\right]\right). \quad (17)$$

The vector inside $\text{Multinomial}(\cdot)$ might not sum to one. In this case, the probability of a class is the ratio of the value in the sum of the vector.

**Posterior of $C_\alpha$ Coordinates**   The generative diffusion kernel $p\left(\mathbf{x}_j^{t-1}\middle|\mathcal{R}^t, \mathcal{C}\right)$ (Eq.7) should align to the posterior obtained from Eq.5 and Eq.6 [Ho et al., 2020]:

$$q(\mathbf{x}_j^{t-1} \mid \mathbf{x}_j^t, \mathbf{x}_j^0) = \mathcal{N}\left(\mathbf{x}_j^{t-1}\middle|\boldsymbol{\mu}_q\left(\mathbf{x}_j^t, \mathbf{x}_j^0\right), \frac{(1 - \bar{\alpha}_{\text{pos}}^{t-1})\beta_{\text{pos}}^t}{1 - \bar{\alpha}_{\text{pos}}^t}\boldsymbol{I}\right), \quad (18)$$

$$\text{where} \quad \boldsymbol{\mu}_q(\cdots) = \frac{\sqrt{\bar{\alpha}_{\text{pos}}^{t-1}}\beta_{\text{pos}}^t}{1 - \bar{\alpha}_{\text{pos}}^{t-1}}\mathbf{x}_j^0 + \frac{\sqrt{\alpha_{\text{pos}}^t}(1 - \bar{\alpha}_{\text{pos}}^{t-1})}{1 - \bar{\alpha}_{\text{pos}}^t}\mathbf{x}_j^t. \quad (19)$$

### A.2  Amino Acid $C_\alpha$ Position Normalization

As amino acid $C_\alpha$ positions could be arbitrary in the 3D space. We need to normalize them to use the standard normal distribution with zero mean and unit variance as the prior distribution. First, we need to derive the statistics of CDR positions. For each CDR in the SAbDab dataset, we shift the overall structure such that the center point of the two CDR anchors is located in origin. Then, we aggregate $C_\alpha$ positions in the shifted CDRs. Finally, we calculate the mean and standard deviation of them. Before training and inference, we shift the whole structure according to their CDR anchors and further shift and scale the structure according to the pre-calculated mean and standard deviation to obtain the normalized coordinates.

## B  Distributions on SO(3)

### B.1  Preliminary: Axis-Angle Representation of Rotations

Conventionally, a rotation is usually represented by 3 Euler angles $(\alpha, \beta, \gamma)$, which can be interpreted as the composition of counter-clockwise rotations by $\alpha, \beta, \gamma$ about $x, y, z$ axes. However, the Euler representation is unsuitable for defining useful operations and distributions w.r.t. rotations considered in this work. Alternatively, we introduce another rotation representation called *axis-angle representations*. This representation parameterized a rotation with an rotational axis $\boldsymbol{u}$ ($\|\boldsymbol{u}\|_2 = 1$) and an angle $\theta$ ($\theta \in \mathbb{R}$).

### B.2  Logarithm of Rotation Matrices and Exponential of Skew-Symmetric Matrices

**Logarithm of Rotation Matrices**   Derived from the definition of matrix logarithm, the logarithm of a rotation matrix $\boldsymbol{R}$ is a **skew-symmetric matrix** [Gallier and Xu, 2003], which can be represented as:

$$\boldsymbol{S} := \log \boldsymbol{R} = \begin{bmatrix} 0 & -v_z & v_y \\ v_z & 0 & -v_x \\ -v_y & v_x & 0 \end{bmatrix}. \quad (20)$$

It can be proven that $\boldsymbol{v} = [v_x, v_y, v_z]$ is the rotational axis of $\boldsymbol{R}$, and $\|\boldsymbol{v}\|_2$ is the rotational angle. For brevity, we can use the vector notation $\boldsymbol{v}$ to represent a rotation in the logarithm space. The space is also known as $so(3)$ (different from the rotation group SO(3), the symbol is in lowercase).

To efficiently compute the logarithm of a rotation matrix without computing matrix logarithm or solving rotational axis-angle, we can use the following formula [Gallier and Xu, 2003]:

$$\log \boldsymbol{R} = \frac{\theta}{2 \sin \theta} (\boldsymbol{R} - \boldsymbol{R}^\mathsf{T}), \tag{21}$$

where $\theta$ can be obtained from $\theta = \cos^{-1}\left(\frac{\mathrm{Tr}\,\boldsymbol{R}-1}{2}\right)$ by the fact that $\mathrm{Tr}(\boldsymbol{R}) = 1 + 2\cos\theta$. Specially, when $\theta = 0$ (or $\boldsymbol{R} = \boldsymbol{I}$), $\log \boldsymbol{R} = [\boldsymbol{0}, \boldsymbol{0}, \boldsymbol{0}]$.

**Exponential of Skew-Symmetric Matrices**  The inversion of the rotation matrix logarithm is the exponential of skew-symmetric matrices. Derived from the definition of matrix exponential, the conversion formula is [Gallier and Xu, 2003]:

$$\exp \boldsymbol{S} = \boldsymbol{I} + \frac{\sin \|\boldsymbol{v}\|_2}{\|\boldsymbol{v}\|_2} \boldsymbol{S} + \frac{1 - \cos \|\boldsymbol{v}\|_2}{\|\boldsymbol{v}\|_2^2} \boldsymbol{S}^2, \tag{22}$$

where $\boldsymbol{S}$ is a skew-symmetric matrix parameterized by three values $\boldsymbol{v} = [v_x, v_y, v_z]$, identical to the definition in Eq.20.

**Remarks**  The logarithm and exponential defined above provide an easy way to create and manipulate rotations in the axis-angle parameterization space. For example, when we would like to create a rotation matrix with an axis and an angle, we can first create a vector $\boldsymbol{v}$ whose direction is the same as the given axis and whose length equals the angle. Then, we rewrite the vector $\boldsymbol{v}$ into a skew-symmetric matrix $\boldsymbol{S}$, and finally convert it to a rotation matrix by Eq.22. We can also manipulate a rotation matrix, for example, changing its rotational angle, by mapping it to the logarithm space, modifying the skew-symmetric matrix, and finally converting it back to a rotation matrix using the exponential formula.

## B.3  ScaleRot: **Rotation Scaling Function**

When we parameterize a rotation matrix with an axis and an angle, it is natural to define the rotation scaling function ScaleRot as scaling the rotational angle. Formally, the definition is:

$$\mathrm{ScaleRot}(k, \boldsymbol{R}) := \exp\left(k \log \boldsymbol{R}\right), \tag{23}$$

where $k$ is the scaling factor and $\boldsymbol{R}$ is a rotation matrix. Specially, $\mathrm{ScaleRot}(0, \boldsymbol{R}) = \boldsymbol{I}$ for all rotation matrix $\boldsymbol{R}$. Intuitively, scaling a rotation matrix by 0 cancels its effect, leading to identity transformation.

## B.4  $\mathcal{IG}_{\mathrm{SO}(3)}$: **Isotropic Gaussian Distribution on SO(3)**

The isotropic Gaussian distribution on SO(3), denoted as $\mathcal{IG}_{\mathrm{SO}(3)}$, is defined on the axis-angle space of rotation: $\mathbb{S}^2 \times [0, \pi]$, where $\mathbb{S}^2 = \{\|\boldsymbol{x}\|_2 = 1 | \boldsymbol{x} \in \mathbb{R}^3\}$ is the unit sphere in $\mathbb{R}^3$. $\mathcal{IG}_{\mathrm{SO}(3)}$ is parameterized by a mean rotation $\boldsymbol{M}$ and a scalar variance $\sigma^2$. Let $\mathbf{u} \in \mathbb{S}^2$ and $\theta$ denote the rotational axis and angle random variables respectively. We first consider $\mathcal{IG}_{\mathrm{SO}(3)}$ with the identity matrix as its mean: $\mathcal{IG}_{\mathrm{SO}(3)}(\mathbf{u}, \theta | \boldsymbol{I}, \sigma^2)$. Its p.d.f. is defined by the product of the uniform distribution on $\mathbb{S}^2$ and a special angular distribution [Matthies et al., 1970, Nikolayev and Savyolov, 1970, Leach et al., 2022]:

$$p_{\mathcal{IG}_{\mathrm{SO}(3)}}(\mathbf{u}, \theta | \boldsymbol{I}, \sigma^2) = p_{\mathrm{uniform}(\mathbb{S}^2)}(\mathbf{u}) p_{\mathrm{angular}}(\theta | \sigma^2), \tag{24}$$

$$\text{where} \quad p_{\mathrm{uniform}(\mathbb{S}^2)}(\mathbf{u}) = \frac{1}{4\pi} \delta\left(\|\mathbf{u}\|_2 - 1\right), \qquad (\mathbf{u} \in \mathbb{S}^2) \tag{25}$$

$$\text{and} \quad p_{\mathrm{angular}}(\theta | \sigma^2) = \frac{1 - \cos\theta}{\pi} \sum_{l=0}^{\infty} (2l+1) e^{-l(l+1)\sigma^2} \frac{\sin\left(\left(l + \frac{1}{2}\right)\theta\right)}{\sin(\frac{\theta}{2})}. \qquad (\theta \in [0, \pi]) \tag{26}$$

When the mean is other than $\boldsymbol{I}$, to sample from the distribution, we can first sample an rotation $\boldsymbol{E}$ from $\mathcal{IG}_{\mathrm{SO}(3)}(\mathbf{u}, \theta | \boldsymbol{I}, \sigma^2)$. Then, we left-multiply $\boldsymbol{R}$ to $\boldsymbol{E}$ to obtain the desired random value $\boldsymbol{RE}$.

**Sampling** The algorithm for drawing samples from $\mathcal{IG}_{\mathrm{SO}(3)}(\boldsymbol{I}, \sigma^2)$ (here, the mean rotation is identity) can be broken down into two steps.

The first step is to draw a unit vector $\boldsymbol{u}$ from the uniform distribution on $\mathbb{S}^2$, $p_{\mathrm{uniform}(\mathbb{S}^2)}(\boldsymbol{u})$. This can be done efficiently by sampling from the 3D standard Gaussian distribution and then normalizing the sampled vector to unit length.

The second part is drawing samples from $p_{\mathrm{angular}}(\theta|\sigma^2)$, which could be more tricky. We empirically use two different proximate sampling strategies depending on the variance $\sigma^2$. When $\sigma$ is larger than 0.1, the series (Eq.26) converges fast. In such cases, we use histograms to approximate the distribution. In specific, we evenly partition $[0, \pi]$ into 8192 bins, and use the probability density $p_{\mathrm{angular}}(\theta|\sigma^2)$ at the center of each bin as the bin weight. We randomly select a bin according to the weights to draw samples from the discretized distribution. Then, we sample from the uniform distribution spanning from the lower bound to the upper bound of the bin. The discretization process is time-consuming. However, since the variances in the diffusion processes are predetermined, we pre-compute and cache the bins and weights to draw samples efficiently. When $\sigma$ is smaller than 0.1, we approximate the distribution using the truncated Gaussian distribution whose mean is $2\sigma$ and the standard deviation is $\sigma$. Empirically, we find that the above proximate sampling algorithm is sufficient for training and sampling from our diffusion model.

To sample from $\mathcal{IG}_{\mathrm{SO}(3)}$ with an arbitrary mean rotation $\boldsymbol{R}$, we first draw a rotation from $\mathcal{IG}_{\mathrm{SO}(3)}(\boldsymbol{I}, \sigma^2)$, denoted as $\boldsymbol{E}$. Finally, we left-multiply $\boldsymbol{R}$ to $\boldsymbol{E}$ to get the desired sample.

### B.5 Uniform Distribution on SO(3)

The uniform distribution on $\mathrm{SO}(3)$ is equivalent to the uniform distribution of normalized quaternions on $\mathbb{S}^3$ [Shoemake, 1992]. To sample a random rotation uniformly, we first sample a random vector from the 4D standard normal distribution. Next, we normalize the vector and treat it as a quaternion. Finally, we convert the quaternion to a rotation matrix which can be regarded as a sample from the uniform distribution on $\mathrm{SO}(3)$.

## C  Neural Network Parameterization

### C.1  Computing Residue Orientations

The orientation of a residue is determined by the coordinate of its three backbone atoms: $C_\alpha$, $C$, and $N$. Let $\boldsymbol{x}_i^\alpha$, $\boldsymbol{x}_i^C$, and $\boldsymbol{x}_i^N$ denote the 3D coordinates of the three backbone atoms of the $i$-th residue respectively. The orientation of the residue, denoted by $\boldsymbol{O}_i$, can be constructed using the following Gram-Schmidt-based algorithm:

$$\boldsymbol{v}_1 \leftarrow \boldsymbol{x}_i^C - \boldsymbol{x}_i^\alpha, \tag{27}$$

$$\boldsymbol{e}_1 \leftarrow \frac{\boldsymbol{v}_1}{\|\boldsymbol{v}_1\|}, \tag{28}$$

$$\boldsymbol{v}_2 \leftarrow \boldsymbol{x}_i^N - \boldsymbol{x}_i^\alpha, \tag{29}$$

$$\boldsymbol{u}_2 \leftarrow \boldsymbol{v}_2 - \langle \boldsymbol{e}_1, \boldsymbol{v}_2 \rangle \boldsymbol{e}_1, \tag{30}$$

$$\boldsymbol{e}_2 \leftarrow \frac{\boldsymbol{u}_2}{\|\boldsymbol{u}_2\|}, \tag{31}$$

$$\boldsymbol{e}_3 \leftarrow \boldsymbol{e}_1 \times \boldsymbol{e}_2, \tag{32}$$

$$\boldsymbol{O}_i \leftarrow [\boldsymbol{e}_1, \boldsymbol{e}_2, \boldsymbol{e}_3]. \tag{33}$$

### C.2  Architectures
**Amino Acid Embedding Layer** The embedding layer for each amino acid takes into account the following information:

- **Amino acid type**: Each of the 20 amino acid types is represented by an embedding vector denoted by $\boldsymbol{e}_i^{\mathrm{type}}$.

- **Heavy atom local coordinates**: The coordinate of each heavy atom in an amino acid is projected to the local coordinate frame using the rule $\boldsymbol{x}_i^{\mathrm{local}} = \boldsymbol{O}_i^\mathsf{T}(\boldsymbol{x}_i^{\mathrm{atom}} - \boldsymbol{x}_i^\alpha)$. All

of the local coordinates are concatenated into a single vector denoted by $e_i^{\text{coord}}$. If some heavy atoms are missing, their local coordinates are filled with zeros. Note that the local coordinates are invariant to global rotation and translation thanks to the projection rule.

- **Backbone dihedral angles**: The backbone dihedrals of amino acid, including $\phi$, $\psi$, and $\omega$ [Liljas et al., 2016, Ingraham et al., 2019], are transformed using a series of sine and cosine functions with different frequencies, which are then concatenated into a single vector $e_i^{\text{dihed}}$.

- **CDR flags and anchor flags**: Amino acids on the CDR or by the two ends of the CDR (anchors) are differentiated from other amino acids by special 0-1 flags denoted as $e_i^{\text{flag}}$.

All of the vectors above are concatenated and fed to an MLP to produce the final embedding vector for each residue.

**Pairwise Embedding Layer**    Pairwise embeddings include information about the relationship between two residues. The pairwise embedding for residue $i$ and $j$ involves the following information:

- **Amino acid types of both amino acids**: There are $20 \times 20 = 400$ combinations of two amino acid types. We represent each of them using an embedding vector denoted by $z_{ij}^{\text{type}}$.

- **Sequential relative position**: If two residues are on the same chain and their distance on the sequence is less than or equal to 32 ($d_{ij}^{\text{seq}} \in \{-32 \ldots 32\}$), the distance is represented by an embedding vector $z_{ij}^{\text{seq}}$. Otherwise, the distance embedding is filled with zeros.

- **Pairwise distances**: The distances between all pairs of atoms are flattened into a vector and transformed by $e^{-c d_{ij}}$ ($c$ is a learnable coefficient) into the spatial distance embedding $z_{ij}^{\text{dist}}$. Missing pairs are filled with zeros.

- **Pairwise backbone dihedrals**: The backbone dihedrals between any two amino acids $i$ and $j$ are defined as $\phi_{ij} = \text{Dihedral}(\boldsymbol{x}_i^{\text{C}}, \boldsymbol{x}_j^{\text{N}}, \boldsymbol{x}_j^{\alpha}, \boldsymbol{x}_j^{\text{C}})$ and $\psi_{ij} = \text{Dihedral}(\boldsymbol{x}_i^{\text{N}}, \boldsymbol{x}_i^{\alpha}, \boldsymbol{x}_i^{\text{C}}, \boldsymbol{x}_j^{\text{N}})$. These two dihedrals are transformed by a series of sine and cosine functions into pairwise dihedral embeddings $z_{ij}^{\text{dihed}}$.

We concatenate the above vectors and feed them into an MLP to get the final pairwise embeddings for each pair of amino acids $\boldsymbol{z}_{ij}$.

**Encoder**    The encoder for encoding the current diffusion state consists of a stack of orientation-aware invariant 3D attention layers. Its aim is to capture relationships between amino acids and provide high-level representations for each residue to denoise.

Let $\boldsymbol{h}_i^{\ell}$ denote the hidden representation output from the last layer (when $\ell = 0$, the representation is the initial residue embedding). The formulas for computing the logit of attention weight between residue $i$ (query) and $j$ (key) is:

$$a_{ij} = \left\langle \boldsymbol{q}\left(\boldsymbol{h}_i^{\ell}\right), \boldsymbol{k}\left(\boldsymbol{h}_j^{\ell}\right) \right\rangle + f\left(\boldsymbol{z}_{ij}\right) + g\left(\left\{\boldsymbol{O}_i^{\mathsf{T}}(\boldsymbol{x}_j^{\text{atom}} - \boldsymbol{x}_i^{\alpha})\right\}_{\text{atom}}\right), \tag{34}$$

where $\boldsymbol{q}(\cdot)$, $\boldsymbol{k}(\cdot)$, $f(\cdot)$, and $g(\cdot)$ are MLP subnetworks. The attention weights can be obtained by taking softmax: $w_{ij} = \text{softmax}_{j=1}^{N}(a_{ij})$. Note that, for simplicity, we do not consider attention heads in the formula, but in practice we use multiple attention heads and different heads can be combined easily via concatenation.

The formula for computing the value passed from residue $j$ to $i$ is:

$$\boldsymbol{v}_{ij} = \boldsymbol{v}\left(\boldsymbol{h}_j^{\ell}, \boldsymbol{z}_{ij}, \left\{\boldsymbol{O}_i^{\mathsf{T}}(\boldsymbol{x}_j^{\text{atom}} - \boldsymbol{x}_i^{\alpha})\right\}_{\text{atom}}\right), \tag{35}$$

where $\boldsymbol{v}(\cdot)$ is a network consisting of MLPs. Finally, the values along with attention weights are used to update the amino acid representations with residual connection and layer normalization, same as the standard transformer [Vaswani et al., 2017].

### C.3    Notes on the Notations of the Denoising Networks $F$, $G$, and $H$

The notations $F$, $G$, and $H$ do *not only* denote the MLPs following the encoder that outputs denoising results. It refers to the embedding layers, the encoder, and the specific output MLP (for example, $F$ includes the MLP for denoising amino acid types). Therefore, the input to $F$, $G$, and $H$ is the diffusion state (sequence and structure) rather than hidden representations. Treating the three sections as a whole allows us to neatly express the equivariance property of the model.

## C.4   Proof of Equivariance

**Lemma 1.** *The Euclidean distance function between two points is invariant to rotations and translations, i.e.* $d(\boldsymbol{Rx}_1 + \boldsymbol{r}, \boldsymbol{Rx}_2 + \boldsymbol{r}) = d(\boldsymbol{x}_1, \boldsymbol{x}_2), \forall \boldsymbol{R} \in \mathrm{SO}(3), \boldsymbol{r} \in \mathbb{R}^3.$

*Proof.*

$$
\begin{aligned}
d(\boldsymbol{Rx}_1 + \boldsymbol{r}, \boldsymbol{Rx}_2 + \boldsymbol{r}) &= \|(\boldsymbol{Rx}_1 + \boldsymbol{r}) - (\boldsymbol{Rx}_2 + \boldsymbol{r})\|_2 \\
&= \|\boldsymbol{R}(\boldsymbol{x}_1 - \boldsymbol{x}_2)\|_2 \\
&= (\boldsymbol{x}_1 - \boldsymbol{x}_2)^{\mathsf{T}} \boldsymbol{R}^{\mathsf{T}} \boldsymbol{R}(\boldsymbol{x}_1 - \boldsymbol{x}_2) \\
&= \|\boldsymbol{x}_1 - \boldsymbol{x}_2\|_2 \\
&= d(\boldsymbol{x}_1, \boldsymbol{x}_2). \qquad \square
\end{aligned}
$$

**Lemma 2.** *The dihedral function for four points is invariant to rotations and translations, i.e.* $\mathrm{Dihedral}(\boldsymbol{Rx}_1 + \boldsymbol{r}, \boldsymbol{Rx}_2 + \boldsymbol{r}, \boldsymbol{Rx}_3 + \boldsymbol{r}, \boldsymbol{Rx}_4 + \boldsymbol{r}) = \mathrm{Dihedral}(\boldsymbol{x}_1, \boldsymbol{x}_2, \boldsymbol{x}_3, \boldsymbol{x}_4), \forall \boldsymbol{R} \in \mathrm{SO}(3), \boldsymbol{r} \in \mathbb{R}^3.$ *Here,* $\mathrm{Dihedral}(\cdots)$ *is defined as:*

$$
\mathrm{Dihedral}(\boldsymbol{x}_1 \ldots \boldsymbol{x}_4) = \mathrm{atan2}(\boldsymbol{v}_2 \cdot ((\boldsymbol{v}_1 \times \boldsymbol{v}_2) \times (\boldsymbol{v}_2 \times \boldsymbol{v}_3)), \|\boldsymbol{v}_2\| (\boldsymbol{v}_1 \times \boldsymbol{v}_2) \cdot (\boldsymbol{v}_2 \times \boldsymbol{v}_3)), \quad (36)
$$

*where* $\boldsymbol{v}_i = \boldsymbol{x}_{i+1} - \boldsymbol{x}_i$ *(i = 1, 2, 3).*

*Proof.* First, we note that:

$$
(\boldsymbol{Rx}_{i+1} + \boldsymbol{r}) - (\boldsymbol{Rx}_i + \boldsymbol{r}) = \boldsymbol{R}(\boldsymbol{x}_{i+1} - \boldsymbol{x}_i) = \boldsymbol{Rv}_i.
$$

By the equivariance of cross product ($\boldsymbol{Ra} \times \boldsymbol{Rb} = \boldsymbol{R}(\boldsymbol{a} \times \boldsymbol{b})$) and the invariance of inner product ($\boldsymbol{Ra} \cdot \boldsymbol{Rb} = \boldsymbol{a} \cdot \boldsymbol{b}$), we have:

$$
\begin{aligned}
\mathrm{Dihedral}(\boldsymbol{Rx}_i + \boldsymbol{r}|i = 1 \ldots 4) &= \mathrm{atan2}(\boldsymbol{Rv}_2 \cdot (\boldsymbol{R}(\boldsymbol{v}_1 \times \boldsymbol{v}_2) \times \boldsymbol{R}(\boldsymbol{v}_2 \times \boldsymbol{v}_3)), \\
&\qquad \|\boldsymbol{Rv}_2\| \boldsymbol{R}(\boldsymbol{v}_1 \times \boldsymbol{v}_2) \cdot \boldsymbol{R}(\boldsymbol{v}_2 \times \boldsymbol{v}_3)) \\
&= \mathrm{atan2}(\boldsymbol{Rv}_2 \cdot \boldsymbol{R}((\boldsymbol{v}_1 \times \boldsymbol{v}_2) \times (\boldsymbol{v}_2 \times \boldsymbol{v}_3)), \\
&\qquad \|\boldsymbol{v}_2\| (\boldsymbol{v}_1 \times \boldsymbol{v}_2) \cdot (\boldsymbol{v}_2 \times \boldsymbol{v}_3)) \\
&= \mathrm{atan2}(\boldsymbol{v}_2 \cdot ((\boldsymbol{v}_1 \times \boldsymbol{v}_2) \times (\boldsymbol{v}_2 \times \boldsymbol{v}_3)), \\
&\qquad \|\boldsymbol{v}_2\| (\boldsymbol{v}_1 \times \boldsymbol{v}_2) \cdot (\boldsymbol{v}_2 \times \boldsymbol{v}_3)) \\
&= \mathrm{Dihedral}(\boldsymbol{x}_i|i = 1 \ldots 4) \qquad \square
\end{aligned}
$$

**Lemma 3.** *The per-amino-acid orientation* $\boldsymbol{O}_i$ *is equivariant to rotations and translations, i.e.,* $\boldsymbol{O}(\boldsymbol{Rx}_i^\alpha + \boldsymbol{r}, \boldsymbol{Rx}_i^{\mathrm{C}} + \boldsymbol{r}, \boldsymbol{Rx}_i^{\mathrm{N}} + \boldsymbol{r}) = \boldsymbol{RO}(\boldsymbol{x}_i^\alpha, \boldsymbol{x}_i^{\mathrm{C}}, \boldsymbol{x}_i^{\mathrm{N}})$

*Proof.* First, we show that the first two basis vectors $\boldsymbol{e}_1$ and $\boldsymbol{e}_2$ are equivariant:

$$
\begin{aligned}
\boldsymbol{e}_1(\boldsymbol{Rx}_i^\alpha + \boldsymbol{r}, \boldsymbol{Rx}_i^{\mathrm{C}} + \boldsymbol{r}) &= \frac{(\boldsymbol{Rx}_i^{\mathrm{C}} + \boldsymbol{r}) - (\boldsymbol{Rx}_i^\alpha + \boldsymbol{r})}{\|(\boldsymbol{Rx}_i^{\mathrm{C}} + \boldsymbol{r}) - (\boldsymbol{Rx}_i^\alpha + \boldsymbol{r})\|} \\
&= \boldsymbol{R} \frac{\boldsymbol{x}_i^{\mathrm{C}} - \boldsymbol{x}_i^\alpha}{\|v\boldsymbol{x}_i^{\mathrm{C}} - \boldsymbol{x}_i^\alpha\|} \\
&= \boldsymbol{R}\boldsymbol{e}_1(\boldsymbol{x}_i^\alpha, \boldsymbol{x}_i^{\mathrm{C}}).
\end{aligned}
$$

Let $\boldsymbol{v}_2 = \boldsymbol{x}_i^{\mathrm{N}} - \boldsymbol{x}_i^\alpha$. We have $(\boldsymbol{Rx}_i^{\mathrm{N}} + \boldsymbol{r}) - (\boldsymbol{Rx}_i^\alpha + \boldsymbol{r}) = \boldsymbol{Rv}_2$. Then, we can prove the equivariance of $\boldsymbol{e}_2$:

$$
\begin{aligned}
\boldsymbol{e}_2(\boldsymbol{Rx}_i^\alpha + \boldsymbol{r}, \boldsymbol{Rx}_i^{\mathrm{C}} + \boldsymbol{r}, \boldsymbol{Rx}_i^{\mathrm{N}} + \boldsymbol{r}) &= \boldsymbol{Rv}_2 - \langle \boldsymbol{Re}_1, \boldsymbol{Rv}_2 \rangle \boldsymbol{Re}_1 \\
&= \boldsymbol{Rv}_2 - \langle \boldsymbol{e}_1, \boldsymbol{v}_2 \rangle \boldsymbol{Re}_1 \\
&= \boldsymbol{R}(\boldsymbol{v}_2 - \langle \boldsymbol{e}_1, \boldsymbol{v}_2 \rangle \boldsymbol{e}_1) \\
&= \boldsymbol{Re}_2(\boldsymbol{x}_i^\alpha, \boldsymbol{x}_i^{\mathrm{C}}, \boldsymbol{x}_i^{\mathrm{N}})
\end{aligned}
$$

By the equivariance of cross product, it is straightforward to show that $\boldsymbol{e}_3$ is also equivariant. Finally, combining the equivariance of $\boldsymbol{e}_1$, $\boldsymbol{e}_1$, and $\boldsymbol{e}_3$, we prove the equivariance of the orientation matrix:

$$
\begin{aligned}
\boldsymbol{O}(\boldsymbol{Rx}_i^\alpha + \boldsymbol{r}, \boldsymbol{Rx}_i^{\mathrm{C}} + \boldsymbol{r}, \boldsymbol{Rx}_i^{\mathrm{N}} + \boldsymbol{r}) &= [\boldsymbol{Re}_1, \boldsymbol{Re}_2, \boldsymbol{Re}_3] \\
&= \boldsymbol{RO}(\boldsymbol{x}_i^\alpha, \boldsymbol{x}_i^{\mathrm{C}}, \boldsymbol{x}_i^{\mathrm{N}}). \qquad \square
\end{aligned}
$$

**Lemma 4.** *The per-amino-acid and pairwise embedding layers are invariant to rotations and translations of the input structure. i.e.*

$$e(s_i, \{\boldsymbol{x}_i^{atom}\}_{atom}, \phi_i, \psi_i, \omega_i, e_i^{flag}) = e(s_i, \{\boldsymbol{R}\boldsymbol{x}_i^{atom} + \boldsymbol{r}\}_{atom}, \phi_i, \psi_i, \omega_i, e_i^{flag}), \quad and$$

$$\boldsymbol{z}(\{d(\boldsymbol{x}_i^{atom1}, \boldsymbol{x}_j^{atom2})\}_{atom1,\ atom2}, \cdots) = \boldsymbol{z}(\{d(\boldsymbol{R}\boldsymbol{x}_i^{atom1} + \boldsymbol{r}, \boldsymbol{R}\boldsymbol{x}_j^{atom2} + \boldsymbol{r})\}_{atom1,\ atom2}, \cdots).$$

*Proof.* Before embedding atom positions for an amino acid, the network first projects the positions using the orientation by the rule:

$$\boldsymbol{x}_i^{\mathrm{local}} = \boldsymbol{O}_i^{\mathsf{T}}(\boldsymbol{x}_i^{\mathrm{atom}} - \boldsymbol{x}_i^{\alpha})$$

The projection operation is invariant to rotations and translations, using Lemma 3:

$$\begin{aligned}
\boldsymbol{x}_i^{\mathrm{local}}(\boldsymbol{R}\boldsymbol{x}_i^{\mathrm{atom}} + \boldsymbol{r}, \boldsymbol{R}\boldsymbol{x}_i^{\alpha} + \boldsymbol{r}) &= (\boldsymbol{R}\boldsymbol{O}_i)^{\mathsf{T}}((\boldsymbol{R}\boldsymbol{x}_i^{\mathrm{atom}} + \boldsymbol{r}) - (\boldsymbol{R}\boldsymbol{x}_i^{\alpha} + \boldsymbol{r})) \\
&= \boldsymbol{O}_i^{\mathsf{T}}\boldsymbol{R}^{\mathsf{T}}\boldsymbol{R}(\boldsymbol{x}_i^{\mathrm{atom}} - \boldsymbol{x}_i^{\alpha}) \\
&= \boldsymbol{x}_i^{\mathrm{local}}(\boldsymbol{x}_i^{\mathrm{atom}}, \boldsymbol{x}_i^{\alpha}).
\end{aligned}$$

The formulas for computing dihedral angles ($\phi_i, \psi_i, \omega_i$) are also invariant by Lemma 2 Other variables (amino acid types and CDR flags) are independent of the 3D structure and hence they are invariant.

So far, we have shown that all the components of embedding layers are invariant to rotations and translations of the overall 3D structure. Therefore, the embedding layer is invariant.

Pairwise embedding layers involve distances between residues, which are invariant by Lemma 2. Other variables are irrelevant to 3D structures. Hence, the pairwise embedding layer is invariant. $\quad\square$

**Lemma 5.** *The orientation-aware attention layer is invariant to rotations and translations if the input hidden representations $\boldsymbol{h}_i, \boldsymbol{z}_{ij}(i, j = 1 \ldots N)$ come from invariant functions.*

*Proof.* First, we show that projecting atoms on the $j$-th amino acid to the orientation of the $i$-th amino acid is invariant to rotations and translations by Lemma 3:

$$(\boldsymbol{R}\boldsymbol{O}_i)^{\mathsf{T}}((\boldsymbol{R}\boldsymbol{x}_j^{\mathrm{atom}} + \boldsymbol{r}) - (\boldsymbol{R}\boldsymbol{x}_i^{\alpha} + \boldsymbol{r})) = \boldsymbol{O}_i^{\mathsf{T}}\boldsymbol{R}^{\mathsf{T}}\boldsymbol{R}(\boldsymbol{x}_j^{\mathrm{atom}} - \boldsymbol{x}_i^{\alpha}).$$

As other inputs to the attention layer ($\boldsymbol{h}_i, \boldsymbol{z}_{ij}(i, j = 1 \ldots N)$) are invariant to rigid transforms on the structure, the networks for computing attention weights and values are invariant. Hence, the attention layer is invariant.

In the case where we stack multiple attention layers, each layer outputs invariant representations for its next layer. Therefore, the network consisting of multiple attention layers is invariant. $\quad\square$

**Proposition 2.** *For any proper rotation matrix $\boldsymbol{R} \in \mathrm{SO}(3)$ and any 3D vector $\boldsymbol{r} \in \mathbb{R}^3$ (rigid transformation $(\boldsymbol{R}, \boldsymbol{r}) \in \mathrm{SE}(3)$), F, G and H satisfy the following equivariance properties:*

$$F(\boldsymbol{R}\mathcal{R}^t + \boldsymbol{r}, \boldsymbol{R}\mathcal{C} + \boldsymbol{r}) = F(\mathcal{R}^t, \mathcal{C}), \tag{37}$$

$$G(\boldsymbol{R}\mathcal{R}^t + \boldsymbol{r}, \boldsymbol{R}\mathcal{C} + \boldsymbol{r}) = \boldsymbol{R}G(\mathcal{R}^t, \mathcal{C}), \tag{38}$$

$$H(\boldsymbol{R}\mathcal{R}^t + \boldsymbol{r}, \boldsymbol{R}\mathcal{C} + \boldsymbol{r}) = \boldsymbol{R}H(\mathcal{R}^t, \mathcal{C}), \tag{39}$$

*where $\boldsymbol{R}\mathcal{R}^t + \boldsymbol{r} := \{s_j^t, \mathbf{x}_j^t + \boldsymbol{r}, \boldsymbol{R}\boldsymbol{O}_j^t\}_{j=l+1}^{l+m}$ and $\boldsymbol{R}\mathcal{C} + \boldsymbol{r} := \{s_i, \boldsymbol{x}_i + \boldsymbol{r}, \boldsymbol{R}\boldsymbol{O}_i\}_{i \in \{1 \ldots N\} \setminus \{l+1, \ldots, l+m\}}$ denote the transformed structure.*

*Proof.* By Lemma 5, we know that the encoder network produces invariant representations. Therefore, the MLP for predicting amino acid types that transforms the invariant representations into a probability over 20 categories is invariant, so $F$ is invariant.

The MLP for predicting local coordinate changes $\mathrm{MLP}_G(\boldsymbol{h}_i)$ is invariant. The local coordinate change is converted to the global coordinate change using the following rule:

$$\hat{\epsilon}_j = \mathbf{O}_j^t \, \mathrm{MLP}_G(\boldsymbol{h}_j).$$

By Lemma 3, the above rule is equivariant to rotations, and hence $G$ is equivariant to rotations.

Similarly, the MLP for predicting changes in orientation $\mathrm{MLP}_H(\boldsymbol{h}_i)$ is invariant. The changes is applied to the original orientation by:

$$\widehat{\mathbf{O}}_j^{t-1} = \mathbf{O}_j^t \boldsymbol{M}_j,$$

which is equivariant to rotations according to Lemma 3. Therefore, $H$ is equivariant to rotations. $\quad\square$

# D  Sampling Algorithms

## D.1  Backbone Atoms and Sidechain $C_\beta$ Construction

The coordinates of backbone atoms (N, $C_\alpha$, C, O) and sidechain $C_\beta$ can be determined by the orientation and the $C_\alpha$ position of an amino acid because the geometry of these atoms is inflexible [Liljas et al., 2016]. To construct the position of N, $C_\alpha$, C, and $C_\beta$ for the $i$-th amino acid, we use the following formula:

$$\boldsymbol{x}_i^{\text{atom}} = \boldsymbol{O}_i \boldsymbol{c}^{\text{atom}} + \boldsymbol{x}_i, \quad (\text{atom} \in \{\text{N}, \text{C}_\alpha, \text{C}, \text{C}_\beta\}) \tag{40}$$

where $\boldsymbol{O}_i$ and $\boldsymbol{x}_i$ is the model-predicted amino acid orientation and $C_\alpha$ position. $\boldsymbol{c}^{\text{atom}}$ is the local coordinate derived from experimental data relative to the orientation and the $C_\alpha$ position, as shown in the following table.

| Atom | $c_x$ | $c_y$ | $c_z$ |
|------|------|------|------|
| N | -0.526 | 1.361 | 0.000 |
| $C_\alpha$ | 0.000 | 0.000 | 0.000 |
| C | 1.525 | 0.000 | 0.000 |
| $C_\beta$ | -0.500 | -0.733 | -1.154 |

The position of O depends on the $\psi$ angle of the amino acid, which relies on the next amino acid in the sequence. Therefore, after constructing backbone atoms, we calculate the $\psi$ angle for each amino acid ($\psi_i = \text{Dihedral}(\text{N}^i, \text{C}_\alpha^i, \text{C}^i, \text{N}^{i+1})$), and use the following rule to construct O coordinates:

$$\boldsymbol{x}_i^{\text{O}} = \boldsymbol{O}_i \boldsymbol{c}^{\text{O}}(\psi_i) + \boldsymbol{x}_i, \tag{41}$$

where

$$\boldsymbol{c}^{\text{O}}(\psi_i) = \begin{bmatrix} 1 & 0 & 0 \\ 0 & \cos\psi_i & -\sin\psi_i \\ 0 & \sin\psi_i & \cos\psi_i \end{bmatrix} \begin{bmatrix} 2.151 \\ -1.062 \\ 0.000 \end{bmatrix}. \tag{42}$$

## D.2  Sidechain Packing and Full Atom Refinement

We use `PackRotamersMover` in PyRosetta [Chaudhury et al., 2010] to pack sidechains only for amino acids on the generated CDR. The packing program is based on the Dunbrack 2010 rotamer library [Shapovalov and Dunbrack Jr, 2011] and the `REF2015` energy function [Alford et al., 2017].

After packing sidechains, we refine the structure with OpenMM [Eastman et al., 2017]. Specifically, we first use PDBFixer to prepare the structure for refinement. We minimize the potential energy of the structure. The potential energy is AMBER99SB force field plus quadratic constraint terms that restrain the position of atoms outside the generated CDR.

# E  Source Code

Code and data are available at https://github.com/luost26/diffab