# OpenReview forum: "Antigen-Specific Antibody Design and Optimization with Diffusion-Based Generative Models for Protein Structures"
_NeurIPS.cc/2022/Conference — NeurIPS 2022 Accept_

### Official Review · Reviewer_B4br · 2022-06-26

**Rating:** 6
**Confidence:** 4
**Soundness:** 2 fair
**Presentation:** 3 good
**Contribution:** 3 good

**Summary:**

The paper presents a diffusion model for jointly generating the CDR sequence and structure of an antibody conditioned on its framework regions and a target antigen. Unlike existing methods, the presented method enables i) conditioning the generation on the antigen structure instead of only framework regions, and ii) predicting side-chain orientations. Lastly, the paper also presents a dataset of experimentally validated antibody structures iii) augmented by pseudo-antibody structures derived from the structure of non-antibody proteins.

**Questions:**

## Questions
1) Does the diffusion model perform better than alternative models such as an autoregressive model (e.g. Jin et. al) or Transformer? Can you show this by ablation? Only change the architecture while using the same (augmented) dataset for training and tuning hyper-parameters appropriately.

2) How does the proposed method compare to a simple sequence-based, non-iterative, model that generates CDR regions conditioned on the sequence (not the structure) of framework regions and the antigen, and predicting the structure with AlphaFold?

3) Please compare to Jin et. al (http://arxiv.org/abs/2110.04624) also in section 4.2 and section 4.4. by training the model on the same augmented dataset of tuning hyper-parameters appropriately.

4) Please show by ablation the effect of i) conditioning on framework regions and ii) the antigen structure.

5) What is the training and inference time of the diffusion model compared to an autoregressive model (e.g. Jin et. al)?

6) How does the model scale to long sequences, which is important for generating long CDRs or also framework regions?

7) Does augmenting the training dataset by pseudo-structures improve the performance of the diffusion model and baseline model? You are referring to results in the appendix. However, I did not find any results. Since this is a major contribution of your paper, results should be in the main text.

8) What is the accuracy of predicting side-chain orientations before refining them by Rosetta?

9) What is the influence of applying the Rosetta packing algorithm and AMBER on the final structure? Does it change the ranking of methods?

10) How do you apply the method to double-stranded antibodies? Do you generate the two different chains independently?

11) Why does IMP% in Table 3 decrease with increasing t?

12) What is the message of figure 3 and figure 4? Consider showing these figures in the appendix. Otherwise, interpret samples and also show samples of baseline methods.

**Limitations:**

Yes

**Strengths And Weaknesses:**

## Strengths
* The paper is mostly clearly written
* The use of a diffusion-based model for generating antibodies is new
* The application of antibody design is important, e.g. for drug development

## Weaknesses
* The benefit of the diffusion-model over alternative generative models is not evaluated experimentally
* The accuracy of predicted side-chain orientations without refinement by the Rosetta packing algorithm is not evaluated experimentally
* The benefit of augmenting the dataset by pseudo-antibody structures is not evaluated experimentally
* Important baselines are missing

---

> ### Author Response · Authors · 2022-08-02
> **Response to Q1**
>
> We thank the reviewer for the valuable comments and below is our response to the questions.
>
> ----
>
> **[Q1] Comparison to transformer and auto-regressive baselines.**
>
> As suggested by the reviewer, we set up two additional baselines for comparison: (1) the first baseline *autoregressively* generates CDR sequences and predicts structures simultaneously; (2) the second baseline generates sequences and structures in one-shot, without iterative processes. These two baseline models share the same transformer-based network architecture except for their output layers. They are trained on the same augmented training dataset. More details about the baselines and experimental results are put to **Section E.4** in the updated version of the supplementary material.
>
> Clearly, our proposed model achieves better scores than the baselines. We think it is not surprising for the following reasons:
> Autoregressive models generate residue sequentially and shortsightedly. That means a residue generated halfway might limit the quality of the final generation result..
> As for the transformer-based model that generates sequences and structures in one-shot, it is unlikely that the network generates a perfect structure within only one step.
> In contrast to these two methods, diffusion models generate structures by iteratively refining them. The iterative refinement process can not only eliminate flaws gradually, but also attend to global information. This is what autoregressive models and one-shot models cannot do and makes diffusion-based models more competitive.
>
> We would like to comment more on why diffusion-based models are more suitable for this task by referring to the recent advances in machine learning for biology and chemistry.
> One of the core ingredients of diffusion-based models is its iterative refinement process. This iterative principle has been shown a more effective way to generate biological and chemical 3D structures, e.g. 3D molecules [1,2], and molecular conformations [3]. The most prominent one is AlphaFold2, whose structure module generates protein structures by repeatedly refining them.
> Different from image or text generation, molecular structure generation demands high precision. For example, a 10 unit deviation in the RGB-color space might not affect human perception of an image. However, a 1Å deviation in molecules might lead to significant change. Therefore, iterative approaches including diffusion models that excel at eliminating flaws and considering global information are a better choice.
>
> [1] Hoogeboom, Emiel, et al. Equivariant diffusion for molecule generation in 3d. International Conference on Machine Learning. PMLR, 2022.
>
> [2] Satorras, Victor Garcia, et al. E (n) Equivariant Normalizing Flows. Advances in Neural Information Processing Systems. 2021.
>
> [3] Xu, Minkai, et al. Geodiff: A geometric diffusion model for molecular conformation generation. ICLR, 2022.

---

> > ### Author Response · Authors · 2022-08-02
> > **Response to Q2-5**
> >
> > **[Q2] About sequence-based models.**
> >
> > We have considered this option when conducting the experiments to evaluate our model. However, we found the comparison is unfeasible and therefore dropped this option.
> > It has been shown that AlphaFold2 has its limitations. AlphaFold2 can accurately predict single-chain proteins, protein complexes with paired Multiple Sequence Alignments (MSAs) [4] and a portion of protein complexes without paired MSAs [5]. However, it generally struggles to predict antibody-antigen complexes [6]. The underlying reason is that AlphaFold2 has to rely on the coevolution information provided by MSAs. However, antibodies are produced by B-cells in response to the antigenic stimulations, which have no homologous sequences in any other species.
> > Therefore, if we predict antibody sequences based on an antigen sequence, it is most likely that we cannot get a reasonable structure of the antibody-antigen complex. Consequently, we can hardly estimate binding energies and RMSDs that rely on the predicted antibody-antigen structures.
> >
> > The reviewer might think that we can still use a neutralization predictor like Jin et al. to evaluate generation quality. However, a neutralization predictor is only applicable for a narrow class of antigens (e.g. SARS CoV) and training the predictor requires a lot of known antibody sequences that are effective to the antigen. Requiring a lot of known antibodies would significantly limit the ability to design antibodies for new antigen targets that have only a few known effective antibodies. In general, this is not transferable to other antigens and clearly deviates from the goal of our work: designing antibodies for an arbitrarily given antigen structure.
> >
> > [4] Evans, Richard, et al. Protein complex prediction with AlphaFold-Multimer. BioRxiv (2021).
> >
> > [5] Gao, Mu, et al. AF2Complex predicts direct physical interactions in multimeric proteins with deep learning. Nature communications 13.1 (2022): 1-13.
> >
> > [6] Yin, Rui, et al. Benchmarking AlphaFold for protein complex modeling reveals accuracy determinants. Protein Science 31.8 (2022): e4379.
> >
> > ----
> >
> > **[Q3] About the model by Jin et al.**
> >
> > We find that our model and the model by Jin et al. are hard to compare directly.
> > The goal of our model is to design antibodies for an arbitrarily given antigen structure. To achieve this goal, our model learns how residues interact with each other in the 3D space and generates CDR residues that fit (interact with) the antigen surface directly in the 3D space.
> > In contrast, the model by Jin et al. does not generate CDRs that fit a specific antigen structure. It mainly learns the sequences and structures of antibody CDR alone. It can target a specific antigen but this is done via a non-transferable sequence-based neutralization predictor, rather than learning how amino acids interact with each other in the 3D space.
> > The neutralization predictor is not transferable because it requires a lot of known antibody sequences that are effective to the antigen to train on. When there are a few known antibody sequences, which is often the case, we can hardly obtain a reliable predictor.
> > Therefore, the capability of the model by Jin et al. is limited in this sense and cannot be directly applied to our more general setting.
> >
> > Nonetheless, though the model by Jin et al. is not directly comparable, we still include baselines that share the same methodology and similar architectures with it (**Section E.4** in the supplementary material). They are trained on the same augmented dataset. We believe they are sufficient to show our model’s advantages.
> >
> > ----
> >
> > **[Q4] About conditioning on antigen structures and antibody framework regions.**
> >
> > Our goal is to generate CDRs for an arbitrary antigen structure. Antigen structures are the input in the setting considered in our work. Ablating them leads to a different research topic.
> > The antibody framework regions are required for building a full antibody-antigen complex. If we remove the framework, we would eventually need to graft the CDR back to the framework. Therefore, we chose to kept the antibody framework and directly generate CDRs on it.
> >
> > ----
> >
> > **[Q5] Training and inference time.**
> >
> > The training time of our model and the baseline models are close, as they share the same architecture except for the output layers. They are all trained for 300K iterations. It takes 95h24m for our diffusion-based model, 51h45m for the autoregressive baseline, and 51h33m for the simple transformer baseline that generates CDR in one-shot.
> >
> > The sampling time for a CDR with 10 residues varies across different models. It takes 8.94secs for the diffusion-based model, 0.12secs for the autoregressive baseline, and 0.30secs for the transformer baseline.
> > The diffusion-based model takes most time because the model is run for 100 times to go through the whole diffusion process. The autoregressive model’s inference time is dependent on the CDR length and 10 residues need 10 runs.

---

> > > ### Author Response · Authors · 2022-08-02
> > > **Response to Q6-9**
> > >
> > > **[Q6] Scalability to long sequences.**
> > >
> > > The length of CDR-H3 in therapeutic antibodies mostly ranges from 10 to 15. Our testset contains 5 short (5\~10 residues), 10 medium (11\~15 residues), and 5 long (16\~20 residues) CDR-H3s, which are sufficiently representative for most therapeutic antibodies. As for the other regions of an antibody, they can be considered constant when we design CDRs, and they are not the major factor of antigen-binding. Therefore, we believe it is reasonable to focus on generating only CDRs in our setting.
> > >
> > > ----
> > >
> > > **[Q7] About the augmented training dataset.**
> > >
> > > The results about the performance of models trained with/without the augmented dataset are put to the **Section E.3** in the supplementary material.
> > >
> > > According to the results, our main finding is that the contribution of the augmented dataset is most significant on CDR-H3s.As CDR-H3 is the most variable region, using extra training data would help the model generalize better and leads to better structure accuracy and sequence recovery.Other CDR regions are conservative, so learning only from antibody structures is somewhat sufficient to model these non-versatile regions.
> > >
> > > ----
> > >
> > > **[Q8] Accuracy of sidechain orientations.**
> > >
> > > Our diffusion model does not explicitly generate sidechains. It generates the orientation of amino acids which determines the direction to which the sidechain stretches.
> > > To evaluate the accuracy of orientation, for each antibody in the testset, we fix the sequence and sample only the structure using our diffusion model (this is actually the CDR loop structure prediction task). We compare the generated orientations to the ground truth orientations and measure the angle error. The average angle error of the first axis in the orientation matrix is 19.7 degrees, and the average angle error of the second axis is 20.6 degrees.
> > >
> > > Actually, in addition to determining sidechains, the orientations also determine the position of backbone atoms other than $C_\alpha$. Therefore, backbone RMSDs can also reflect the accuracy of orientations in a more intuitive way.
> > > We sample only the structure using our diffusion model with the sequence fixed, and calculate the backbone RMSDs of different CDRs:
> > >
> > > | CDR  | H1    | H2    | H3    | L1    | L2    | L3    |
> > > | ---- | ----- | ----- | ----- | ----- | ----- | ----- |
> > > | RMSD | 1.09Å | 0.69Å | 2.56Å | 1.76Å | 0.60Å | 1.17Å |
> > >
> > > We can see that our model predicts CDR backbone structures well, and this reflects the prediction accuracy of amino acid orientations.
> > >
> > > ----
> > >
> > > **[Q9] Influence of AMBER and Rosetta relax.**
> > >
> > > The purpose of AMBER is to repair imperfect bond lengths and bond angles which would result in unrealistic high energy. For example, the ideal length of the N-C peptide bond is 1.329Å. This is a strict restraint and even small deviation (e.g. 0.1Å) from it would lead to a significant penalty by the energy function. If we don’t apply AMBER to idealize bond geometry, these penalties would dominate the energy value, making the energy value nonsensical. In this case, the ranking results are dominated by how ideal the bonds are rather than the binding between the antibody and the antigen.
> > >
> > > To see this concretely, we compute the binding energies of structures before AMBER-Rosetta relaxation and after relaxation. Before relaxation, the average range (max - min) of binding energies is 3573.38. After relaxation, the average range is 77.15. Clearly, the energy values before relaxation greatly diverge.
> > >
> > > As for Rosetta-relax, it is a standard step in the Rosetta scoring protocol [7]. It adjusts the structure to eliminate clashes and makes it favorable to the Rosetta energy function. It is reported that applying relaxation before scoring is important for getting energy values that correlate well with experimental results [7].
> > >
> > > Note that, we applied the same AMBER-Rosetta protocol for all the models (including our model and baselines) to ensure fair comparison.
> > >
> > > In fact, the AMBER-Rosetta relaxation process does not change the structure much. The average RMSD of CDR between the relaxed and unrelaxed structures is 1.31Å. Specially, for the most variable CDR-H3, the average RMSD of CDR between the relaxed and unrelaxed structures is 1.71Å.
> > >
> > > [7] Conway, Patrick, et al. Relaxation of backbone bond geometry improves protein energy landscape modeling. Protein Science 23.1 (2014): 47-55.

---

> > > > ### Author Response · Authors · 2022-08-02
> > > > **Response to Q10-12 (end)**
> > > >
> > > > **[Q10] Double-stranded antibodies.**
> > > >
> > > > Yes. We would generate them independently in this case.
> > > >
> > > > ----
> > > >
> > > > **[Q11] Why does IMP% in Table 3 decrease with increasing t?**
> > > >
> > > > To optimize an antibody, we first perturb it for $t$ steps and then denoise it for another $t$ steps. Therefore, higher $t$ indicates more aggressive sampling and exploration of a larger space around the initial antibody. Exploring in a larger space leads to a lower probability of finding an antibody that has better binding energy than the original antibody (lower IMP%), and the optimized antibodies are less similar to the original one (higher RMSD and lower SeqID).
> > > >
> > > > ----
> > > >
> > > > **[Q12] About the figure.**
> > > >
> > > > Figure 1 shows the complementarity between the antigen (red) and the generated CDRs (blue).
> > > > In sample 1, the sidechains of the antigen ``embrace’’ the CDR without any clashes. Hence, the CDR geometrically fits the best to the antigen among the 3 samples, and it has the lowest binding energy.
> > > > In sample 2, the CDR fits less tightly than the CDR in sample 1, but it is still a good fit. Therefore, it is binding energy is a bit higher.
> > > > In sample 3, there is a significant gap between the CDR and the antigen, which indicates bad complementarity. Therefore, the binding energy is the highest among the 3 samples.

---

> > > > > ### Comment · Reviewer_B4br · 2022-08-09
> > > > > **Increased my score to “weak accept”.**
> > > > >
> > > > > Dear authors,
> > > > >
> > > > > Thanks for addressing all of my comments! I increased my score to “weak accept”.
> > > > >
> > > > > [Q2] I agree with you that MSA based approaches such as AlphaFold2 or RosettaFold are less accurate for predicting CDR loops due to often insufficient homologs for building a reliable MSA. However, recent approaches such as OmegaFold, Nanonet, or IgFold enabled more accurate predictions and it remains unclear if your approach performs better. I believe that you can strengthen your manuscript by including the discussed baseline approach that first generates the antibody sequence conditioned on the antigen, and then uses OmegaFold or AlphaFold2 to predict the structure.
> > > > >
> > > > > [Q3] Thanks for your description and additional experiment. Please describe the differences between Jin et al and your approach in the related work section and refer to your additional experimental results.
> > > > >
> > > > > [Q4] I agree with you that not conditioning the generation on FWRs would require aligning and grafting generated CDRs on the parent structure as also performed by Jin et al. However, it would reveal if doing so improves the quality of generated CDRs.
> > > > >
> > > > > [Q5/Q6] Please describe your runtime measurements in the manuscript. Also highlight that the performance of your model depends less on the length of FWRs, which can be long.
> > > > >
> > > > > [Q9] Thanks for explaining. I would appreciate if you could summarize the RMSD of the relaxed and unrelaxed structures of each method in a table . This will show how much the structures of each methods are changed by relaxing them.
> > > > >
> > > > > [Q11]: Please describe this in the main text.
> > > > >
> > > > > [Q12]: Please also interpret figures in the main text.

---

### Official Review · Reviewer_pxV3 · 2022-07-03

**Rating:** 7
**Confidence:** 4
**Soundness:** 3 good
**Presentation:** 3 good
**Contribution:** 3 good

**Summary:**

This paper addresses the problem of antigen-conditional antibody design. While there has been a line of work using ML for antibody design, existing work mostly focused on antigen-independent antibody generation, yet antigen-conditional antibody design has been far more common in practical use cases. This paper makes progress on an important problem.

Two main contributions of this paper are:
1) A new curated dataset of "antibody-like" loops interacting with other protein chains, extracted from PDB.
2) A new approach for antigen-conditional antibody design (joint sequence-structure design) based on diffusion models.

**Questions:**

Major questions/suggestions:
1. Among the three components (sequence, Ca, orientation), which one is more important to model? Ablation studies would make the paper a lot stronger.
2. The GNN baseline in all results is antigen-independent? If so it'd be greatly appreciated to mark that (e.g. add a column in tables to say antigen dependent/independent), because otherwise it's misleading to directly compare the two.
3. How many antigens are shared between the train and test splits? It would be important to at least show statistics of antigen overlap (e.g. min RMSD or % overlap) between train and test splits.
4. How would the performance degrade if using HDOCK instead of known complexes for all the main results (sections 4.2-4.4)?
5. Comparison to antibody structure prediction methods (e.g. IgFold) for RMSD?

Other questions:
1. Does "side chains" mean "Cb" specifically in the paper? It would be helpful to clarify whether it's all atoms in side chains or just Cb.
2. Compared to using equal probabilities for all AAs in Eq. 1, would it be helpful to use e.g. BLOSUM or other priors?
3. How is the posterior derived from Eq. 1?
4. Is minimizing the expected KL (Eq. 4) equivalent to maximum likelihood estimation (MLE)?
5. It'd be helpful to include simpler baselines, e.g. IMGT alignment consensus of antigen-specific antibodies. Although I expect simpler baselines to be worse, it'd still be helpful to put results into context.
6. It would be more convincing for antibody optimization to also show results on antibody DMS datasets -- unless it's hard to evaluate likelihood in diffusion models?

**Limitations:**

See above ("weaknesses") for limitations. The authors did acknowledge the main limitations and I appreciate that as a reader.

**Strengths And Weaknesses:**

Strengths:
- (Originality & Significance) This paper considers antibody design conditioned on the antigen 3D structure. This has been an important yet challenging task, and this is one of the first ML approaches to do so.
- (Originality) The authors curated a new dataset of antibody-like complexes (loops in contact with another chain). I believe this dataset will be useful for the research community.
- (Clarity) This paper is mostly clearly written with high quality figures, although the writing could be improved at places (e.g. typos and more accurate language about CDRs).
- (Quality) The empirical results seem sound, demonstrating improvements in three different antibody design tasks.

Weaknesses:
- (Significance) While the paper takes a step forward in antigen-conditioned antibody design, there is still a notable gap between the scenario in the paper and the real use case: In a real use case, we do not know the relative orientation between the antibody and the antigen, while in the design tasks in the paper the orientation is given based on known antigen-antibody complexes. The paper does address this concern in Section 4.5 by using HDOCK to dock the antibody template as a precursor step. However, this is only done for one antigen. in my view, this paper would be a lot stronger if the authors can show the design performance in Sections 4.2-4.4 both with and without known bound complexes.
- (Quality) This paper would also be strengthened by more ablation studies on the relative merits of different model components, as well as by the inclusion of simpler baselines.
- (Quality) The split in this paper is based on CDR sequence identity, and not based on antigen structures. It would be helpful to at least show statistics of antigen overlap (e.g. RMSD) between train and test splits.

---

> ### Author Response · Authors · 2022-08-02
> **Response (1/2)**
>
> We thank the reviewer for the valuable comments and below is our response to the questions.
>
> ----
>
> **[Q] About general cases where bound complexes are unknown.**
>
> We used HDOCK to generate antibody orientations, but we found that HDOCK does not always give good predictions. HDOCK outputs dozens of docking structures but a large portion of them are incorrect, e.g. non-paratope regions are bound to the antigen. Therefore, we have to manually select reasonable structures and this makes it difficult to conduct large scale evaluation.
> We also considered using ClusPro but the server does not allow submitting a large bulk of jobs in a short time, and it is much slower than HDOCK due to the server’s long job queue. For AlphaFold2, it has been shown that it struggles to predict antibody-antigen complex structures [1], so it is not an ideal choice for generating antibody-antigen template, either. Therefore, we believe generating antibody orientations for antigens is a challenging problem which might be a good direction for future work, and future efforts would eventually fill the gap in a neat way.
>
> Although we did not fully address the cases where bound complexes are unknown, our main setting where bound complexes are known are practical in some real scenarios.
> Consider a case where we have an antibody-antigen complex structure and we would like to optimize the binding affinity. If mutating several residues on CDRs does not produce desirable results, we could use the proposed model to aggressively redesign the CDRs.
>
> [1] Yin, Rui, et al. Benchmarking AlphaFold for protein complex modeling reveals accuracy determinants. Protein Science 31.8 (2022): e4379.
>
> ----
>
> **[Q] About additional baselines.**
>
> We have conducted experiments with two more baselines:
>
> (1) the first baseline *autoregressively* generates CDR sequences and predicts structures simultaneously; (2) the second baseline generates sequences and structures in one-shot, without iterative processes. These two baseline models share the same transformer-based network architecture except for their output layers. They are trained on the same augmented training dataset. More *details* about the baselines and experimental *results* are put to **Section E.4** in the updated version of the supplementary material.
>
> Clearly, our proposed model achieves better scores than the baselines. We think it is not surprising for the following reasons: Autoregressive models generate residue sequentially and shortsightedly. That means a residue generated halfway might limit the quality of the final generation result. As for the transformer-based model that generates sequences and structures in one-shot, it is unlikely that the network generates a perfect structure within only one step. In contrast to these two methods, diffusion models generate structures by iteratively refining them. The iterative refinement process can not only eliminate flaws gradually, but also attend to global information. This is what autoregressive models and one-shot models cannot do and makes diffusion-based models more competitive.
>
> ----
>
> **[Q] About antigen overlap between train and test splits.**
>
> The testset contains 20 antibody-antigen complexes. 5 of them do not have antigens that are similar (sequence identity >= 50%) to any antigen in the training set.
> Although the remaining 15 complexes contain antigens that appear in the training set, their binding interfaces are different from the training sets. This thanks to that we clustered antibodies according to CDR similarity. Consequently, two antibodies from different clusters bind to the same antigen at different regions and poses.
>
> We quantify the difference in binding interfaces between the training set and test set by the following method: For each antigen in the test set, we find all the similar antigens (sequence identity >= 50%) in the training set. For each antigen pair (an antigen from the testset, and a similar antigen from the training set), we identify residues interacting to the antibody on the two antigens according to their antibody-antigen complexes. Then, we align the two antigens and compute the ratio of common interacting residues as the binding interface similarity.
> The average binding interface similarity between the testset and the training set is 17.6%, and the max similarity is 61.0%. Therefore, we believe our training set and test-set are split properly in terms of antigen overlaps.
>
> (1/2)

---

> > ### Author Response · Authors · 2022-08-02
> > **Response (2/2)**
> >
> > **[Q] About the contribution of the three components (sequence, Ca, orientation).**
> >
> > For the sequence-structure codesign task, the three components are equally important because we need to model both sequence and structure ($C_\alpha$ & orientation).
> > If we fix the sequence and only generate structures ($C_\alpha$ and orientation), the problem becomes CDR structure prediction (loop modeling). Fixing the structure ($C_\alpha$ and orientation) and generating sequences is fix-backbone protein design.
> >
> > Representing the structure using only $C_\alpha$ coordinates is not sufficient. The orientation is used to implicitly represent the coordinate of other backbone atoms including N, C, and O. Therefore, we think both $C_\alpha$ and orientation are indispensable for generating structures.
> >
> > ----
> >
> > **[Q] About the GNN baseline.**
> >
> > The GNN-baseline is antigen-independent. It directly conditions on an arbitrarily given antigen structure. It only shares a similar methodology with the model by Jin et al. We have added more details about the GNN baseline in the **Section E.4** of the updated supplementary material.
> >
> > ----
> >
> > **[Q] Comparison to IgFold.**
> >
> > We find it very hard to compare our model to IgFold mainly for the following reasons: First, IgFold predict antibody structures based on known sequences, but in our setting, antibody sequences are unknown and need generation. Second, IgFold can only predict the structure of antibody alone, rather than antibody-antigen complexes. However, we consider how to design an antibody that binds to the antigen.
> >
> > (2/2)

---

> > > ### Comment · Reviewer_pxV3 · 2022-08-09
> > > **Updated review**
> > >
> > > Dear authors,
> > >
> > > Thank you for taking the time to provide additional information in the author response. This has addressed some of my main concerns, and I have updated the review accordingly to increase the score. Please do include the additional information here (especially about limitations of docking algorithms, antigen overlap, and the importance of the orientation) into the final version of the paper -- I believe the additional information will help clarify the manuscript.

---

### Official Review · Reviewer_tyV1 · 2022-07-12

**Rating:** 7
**Confidence:** 4
**Soundness:** 4 excellent
**Presentation:** 4 excellent
**Contribution:** 4 excellent

**Summary:**

This paper introduces a diffusion-based model for jointly modeling the sequence and structure of complementarity-determining regions (CDRs) of antibodies, conditioned on antigen structure and antibody framework. The method enables the design of CDRs in a wide range of scenarios (from least constrained to most): CDR sequence-structure co-design, fixed-backbone sequence design, or antibody optimization (starting from a known antibody). The approach deviates from prior work notably via the conditioning on the antigen 3D structure and factoring in the side-chain orientations. Experiments demonstrate superior performance in the aforementioned design settings. Lastly, authors curate an extended set of protein structures for model training: antibody-antigen from SAbDab and pseudo-CDRs from the PDB (assimilating loops and the corresponding chains they interact with as proxy antibody-antigen structures).

**Questions:**

Lines 92-93 "It relies on an additional antigen-specific predictor to predict the neutralization of the designed antibodies, which is hard to generalize to arbitrary antigens". Could you please clarify what you meant here regarding difficulty to generalize Vs your method? Do you mean this approach requires to train a new predictor for each new antigen (Vs your method handles any antigen by construction)? If so, isn't the antigen-specific predictor more likely to provide good results in the focus application it's been design for? Or do you have evidence the more general modeling you are proposing is both more flexible and has higher task performance?

Lines 274-275 -- "We find that using the augmented training dataset could enhance the performance on the test set". Did you train a model without the pseudo-CDR structures to give a sense of how useful this data extension is?

Tables 1/2: do you have a sense for the relative contributions to performance lift from the different modeling decisions you made over prior baselines (eg., conditioning on antigen structure, diffusion-based modeling, factoring in side-chain orientations, rotation & translation equivariance)

Minor points:
- Line 513 -- dead reference

**Limitations:**

- Limitations are very briefly discussed in the conclusion.
- Potential negative societal impact of the work is not discussed (developed method may be re-purposed more broadly for other protein design tasks which may be malignant).

**Strengths And Weaknesses:**

**Strengths**
- Significance: the ability to design novel antibodies that optimally bind to target antigens, while satisfying other constraints is critical to various antibody therapies
- Originality: this work makes solid contributions in proposing a model that 1) handles certain design settings more naturally (eg, antibody optimization) 2) performs better than existing baselines (based on experiments reported in this work) 3) models the entire CDR structures with atomic precision through the incorporation of side-chain geometry.
- Clarity: the paper is very well written and structured. Figures and notations are very clear.
- Quality: sound and thorough experimental design.

**Weaknesses**
- Quality: a few minor claims / points are not fully substantiated (see Questions)

---

> ### Author Response · Authors · 2022-08-02
> **Response**
>
> We thank the reviewer for the valuable comments and below is our response to the questions.
>
> ----
>
> **[Q1] About neutralization predictors.**
>
> Yes. A neutralization predictor is trained for some specific class of antigens (e.g. SARS CoV). It requires a lot of known antibody sequences that target the antigen. When we do not have many known antibody sequences for a new antigen, which is often the case, we can hardly obtain a reliable neutralization predictor. Therefore, it is not suitable for antigens that have only a few known effective antibodies.
>
> Another drawback of neutralization predictors is that it is a black-box. It regresses antibody sequences on neutralization or binding affinities to a specific antigen using neural networks. Therefore, it does not have explicit knowledge about how the amino acids on the antibody and antigen interact with each other in the 3D space, which is fundamental to the antigen-binding behavior of antibodies. Lacking interpretability undermines the reliability of neutralization predictors.
>
> In contrast, our model is *structure-based*. It learns to generate amino acids that interact with the amino acids on the given structure (antigen). This is driven by the general rule of how amino acids interact physically, which is independent of antigen types. Therefore, our model is generalizable to arbitrary antigens in this sense.
>
> In addition, our model generates CDRs bound to the antigen in the 3D space. This enables structural (RMSD) and biophysical (binding energy) analysis, interpretation, and evaluation which requires 3D structures. However, a black-box sequence-based model does not provide such interpretability.
>
>
> ----
>
> **[Q2] About the effect of augmented datasets.**
>
> The results about the performance of models trained with/without the augmented dataset are put to the **Section E.3** in the supplementary material.
>
> According to the results, our main finding is that the contribution of the augmented dataset is most significant on CDR-H3s.
> As CDR-H3 is the most variable region, using extra training data would help the model generalize better and lead to better structure accuracy and sequence recovery.
> Other CDR regions are conservative, so learning only from antibody structures is somewhat sufficient to model these non-versatile regions.
>
> ----
>
> **[Q3] About the contribution of different components.**
>
> As discussed in our response to Question 1, our model is structure-based. To achieve the goal of generalizability to arbitrary antigens, we chose to directly condition our model on antigen structures. Removing antigen structures would lead to another setting.
>
> To reveal the contribution of our model design, we have conducted experiments with two more baselines:
>
> (1) the first baseline *autoregressively* generates CDR sequences and predicts structures simultaneously; (2) the second baseline generates sequences and structures in one-shot, without iterative processes. These two baseline models share the same transformer-based network architecture except for their output layers. They are trained on the same augmented training dataset. More *details* about the baselines and experimental *results* are put to **Section E.4** in the updated version of the supplementary material.
>
> Clearly, our proposed model achieves better scores than the baselines. We think it is not surprising for the following reasons: Autoregressive models generate residue sequentially and shortsightedly. That means a residue generated halfway might limit the quality of the final generation result. As for the transformer-based model that generates sequences and structures in one-shot, it is unlikely that the network generates a perfect structure within only one step. In contrast to these two methods, diffusion models generate structures by iteratively refining them. The iterative refinement process can not only eliminate flaws gradually, but also attend to global information. This is what autoregressive models and one-shot models cannot do and makes diffusion-based models more competitive.

---

> > ### Comment · Reviewer_tyV1 · 2022-08-07
> > **Re: Response**
> >
> > Dear authors,
> >
> > Thank you for the additional clarifications. About the responses you provided:
> >
> > [Q1] Agree with the benefits of the structure-based approach in terms of increased interpretability and advantages in the low data regime. While there are many viruses for which few antigens are available (eg., emerging viruses early in a pandemic, under-studied viruses), there are also several viruses for which there are many antibody sequences available. In these instances, neutralization predictors seem to be very compelling options. It does not take anything away from your contributions, but perhaps something to express a bit more clearly in the text to emphasize the situations in which your method will be most compelling Vs prior alternatives.
> >
> > [Q2] Makes sense - thank you.
> >
> > [Q3] Thank you for running these additional experiments. Could you please add architecture details (eg., number of parameters, layers, attention heads) for the two new baselines? How do they compare to your model in terms of total # of params and training / inference time?

---

> > > ### Author Response · Authors · 2022-08-08
> > > **Thank you for your response**
> > >
> > > We thank the reviewer for the valuable comments that help us imporve our work!
> > >
> > > We have added more architecture details (number of parameters, number of encoder layers, number of attention heads, *etc*.) to  **Section E.5** of the updated supplementary material.
> > >
> > > In summary, the sizes of our model and the baselines are about the same as they share the same encoder architecture. As for the inference time, our model takes longer time because the diffusion process requires running the whole network at every step. Our model also takes slightly longer time to train due to the forward diffusion process at each training step.

---

### Meta-Review · Area_Chair_wdF7 · 2022-08-26

**Recommendation:** Accept
**Confidence:** Certain

**Metareview:**

This is a very exciting and timely paper that eleganlty enables CDR sequence-structure co-design, sequence design given a certain backbone, and antibody optimization.

The reviewers and AC all appreciate the extensive feedback provided by the authors and the additional studies included in the supplements.  We strongly encourage the authors to also incorporate in their manuscript certain points made in their feedback. In particular please include comments to
- contrast the proposed approach with (i) the work "Iterative Refinement Graph Neural Network for Antibody Sequence-Structure Co-design" and (ii)neutralization prediction approaches
- highlight the  limitations of docking algorithms and the pertinent future work direction of generating antibody orientations for antigens
- clarify various points, such as description of Figure 1.

**Award:**

No

---

### Decision · Program_Chairs · 2022-09-14

Accept